# Personalized Federated Learning with Gaussian Processes

**Idan Achituve**
Bar-Ilan University, Israel
`idan.achituve@biu.ac.il`

**Aviv Shamsian**
Bar-Ilan University, Israel
`aviv.shamsian@biu.ac.il`

**Aviv Navon**
Bar-Ilan University, Israel
`aviv.navon@biu.ac.il`

**Gal Chechik**
Bar-Ilan University, Israel
NVIDIA, Isreal
`gal.chechik@biu.ac.il`

**Ethan Fetaya**
Bar-Ilan University, Israel
`ethan.fetaya@biu.ac.il`

## Abstract

Federated learning aims to learn a global model that performs well on client devices with limited cross-client communication. Personalized federated learning (PFL) further extends this setup to handle data heterogeneity between clients by learning personalized models. A key challenge in this setting is to learn effectively across clients even though each client has unique data that is often limited in size. Here we present *pFedGP*, a solution to PFL that is based on Gaussian processes (GPs) with deep kernel learning. GPs are highly expressive models that work well in the low data regime due to their Bayesian nature. However, applying GPs to PFL raises multiple challenges. Mainly, GPs performance depends heavily on access to a good kernel function, and learning a kernel requires a large training set. Therefore, we propose learning a shared kernel function across all clients, parameterized by a neural network, with a personal GP classifier for each client. We further extend pFedGP to include inducing points using two novel methods, the first helps to improve generalization in the low data regime and the second reduces the computational cost. We derive a PAC-Bayes generalization bound on novel clients and empirically show that it gives non-vacuous guarantees. Extensive experiments on standard PFL benchmarks with CIFAR-10, CIFAR-100, and CINIC-10, and on a new setup of learning under input noise show that pFedGP achieves well-calibrated predictions while significantly outperforming baseline methods, reaching up to 21% in accuracy gain.

## 1 Introduction

In recent years, there is a growing interest in applying learning in decentralized systems under the setup of federated learning (FL) [37, 51, 66]. In FL, a server node stores a global model and connects to multiple end-devices ("clients"), which have private data that cannot be shared. The goal is to learn the global model in a communication-efficient manner. However, learning a single shared model across all clients may perform poorly when the data distribution varies significantly across clients. *Personalized Federated Learning* (PFL) [67] addresses this challenge by jointly learning a personalized model for each client. While significant progress had been made in recent years, leading approaches still struggle in realistic scenarios. First, when the amount of data per client is limited, even though this is one of the original motivations behind federated learning [4, 51, 72]. Second, when the input distribution shifts between clients, which is often the case, as clients use different devices and sensors. Last, when we require well-calibrated predictions, which is an important demand from medical and other safety-critical applications.

35th Conference on Neural Information Processing Systems (NeurIPS 2021).

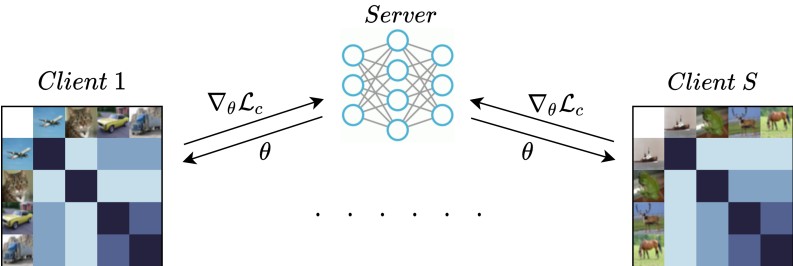

Figure 1: pFedGP - learning a shared deep kernel function with client-specific GP models. Each client stores private data, possibly from a different distribution. The data is first mapped to an embedding space with a shared neural network across all clients. Then, using common kernels a GP is applied to the data of the client for model learning and inference. We illustrate the per-client kernel matrix $k_\theta(\mathbf{x}_i, \mathbf{x}_j)$. Bold cells indicate a stronger covariance.

Here, we show how *Gaussian Processes* (GPs) with deep kernel learning (DKL) [80] is an effective alternative for handling these challenges. GPs have good predictive performance in a wide range of dataset sizes [2, 81], they are robust to input noise [75], can adapt to shifts in the data distribution [48], and provide well-calibrated predictions [69]. While regression tasks are more natural for GPs, here we focus on classification tasks for consistency with common benchmarks and learning procedures in the field; however, our approach is also applicable to regression tasks.

Consider a naive approach that fits a separate GP classifier to each client based on its personal data. Its performance heavily depends on the quality of the kernel, and standard kernels tend to work poorly in domains such as images. A popular solution to this problem is to use deep kernel learning (DKL) [80], where a kernel is applied to features outputted by a neural network (NN). Unfortunately, GPs with DKL can strongly overfit, often even worse than standard NNs [56], and thus negate the main benefit of using a GP. We solve this issue by jointly learning a shared kernel function across clients. As the kernel captures similarities between inputs, a single kernel should work well across clients, while using a separate GP per client will give the required flexibility for personalization.

We adapt a GP classifier recently proposed in [2] which uses the Pólya-Gamma augmentation [57] in a tree-structure model to the federated setting. We term our method *pFedGP*. We extend pFedGP by tailoring two inducing points (IPs) methods [58, 70]. The first helps generalization in the low data regime and, unlike common inducing point methods, does not reduce the computational costs. The second does focus on reducing the computational cost to make our approach scalable and work in low-resource clients. We also adjust previous PAC-Bayes generalization bounds for GPs [60, 64] to include the Pólya-Gamma augmentation scheme. These bounds are suitable for cases where the kernel is not learned, such as when new clients arrive after the shared NN was already learned.

Therefore, this paper makes the following contributions: (i) introduce pFedGP as a natural solution to PFL; (ii) develop two IP methods to enhance GP classifiers that use the Pólya-Gamma augmentation scheme and integrate them with pFedGP; (iii) derive a PAC-Bayes generalization bound on novel clients and show empirically that it gives meaningful guarantees; (iv) achieve state-of-the-art results in a wide array of experiments, improving accuracy by up to 21% [1].

## 2 Related work

**Federated learning.** In FL, clients collaboratively solve a learning task while preserving data privacy and maintaining communication efficiency [1, 34, 42, 51, 54, 82, 85]. FedAvg [51] is an early but effective FL approach that updates models locally and averages them into a global model. Several optimization methods have been proposed for improving convergence in FL [41, 45, 71, 77]. Other approaches focus on preserving client privacy [3, 20, 52, 90], improving robustness to statistical diversity [26, 27, 31, 35, 87, 88], and reducing communication cost [13, 61]. These methods aim to learn a global model across clients, limiting their ability to deal with heterogeneous datasets.

---

[1]Our code is publicly available at https://github.com/IdanAchituve/pFedGP

**Personalized federated learning.** To overcome client heterogeneity, PFL aims to introduce some personalization for each client in the federation [39, 73]. Recent methods include adapting multitask learning [18, 67], meta-learning approaches [6, 21, 22, 33, 43, 89], and model mixing, where clients learn a mixture of the global and local models [4, 17, 27, 44]. Other approaches utilize different regularization schemes to enforce soft parameter sharing [32, 72]. Personalization in FL has also been explored through clustering approaches in which similar clients within the federation have a greater effect on one another [49, 86]. Recently, [65] proposed learning a central hypernetworks that acts on client representation vectors for generating personalized models.

**Bayesian FL.** Some studies put forward a Bayesian treatment to the FL setup. [8, 12] used variational inference with Bayesian NNs. [76, 84] proposed a matching algorithm between local models based on the Beta-Bernoulli process to construct a global model. [14] extended Bayesian optimization to FL setting via Thompson sampling. To scale the GP model they used random Fourier features. We use inducing points instead. [83] proposed a federated learning framework that uses a global GP model for regression tasks and without DKL. Unlike this study, we focus on classification tasks with a personal GP classifier per client and advocate sharing information between clients through the kernel. [74] used GPs in a client selection strategy. In [36] an approach based on stein variational gradient descent was suggested. This method does not scale beyond small-sized networks. [47] proposed a multivariate Gaussian product mechanism to aggregate local models. As we will show, this method is less suited when the data heterogeneity between clients is large.

**Gaussian process classification.** Unlike regression, in classification approximations must be used since the likelihood is not a Gaussian [59]. Classic approaches include the Laplace approximation [79], expectation-propagation [53], and least squares [62]. Recently, several methods were proposed based on the Pólya-Gamma augmentation [57] for modeling multinomial distributions [46], GP classification [23, 24, 78], few-shot learning [69], and incremental learning [2]. Here we build on the last approach. Classification with GPs is commonly done with variational inference techniques [30], here we wish to exploit the conjugacy of the model to take Gibbs samples from the posterior. This approach yields well calibrated [69] and more accurate models [2].

## 3   Gaussian processes background

We first provide a brief introduction to the main components of our model. Detailed explanations are deferred to the Appendix. Scalars are denoted with lower-case letters (e.g., $x$), vectors with bold lower-case letters (e.g., $\mathbf{x}$), and matrices with bold capital letters (e.g., $\mathbf{X}$). In general, $\mathbf{y} = [y_1, ..., y_N]^T$ is the vector of labels, and $\mathbf{X} \in \mathbb{R}^{N \times d}$ is the design matrix with $N$ data points whose $i^{th}$ row is $\mathbf{x}_i$.

**Gaussian processes.** GPs map input points to target output values via a random latent function $f$. $f$ is assumed to follow a Gaussian process prior $f \sim \mathcal{GP}(m(\mathbf{x}), k(\mathbf{x}, \mathbf{x}'))$, where the evaluation vector of $f$ on $\mathbf{X}$, $\mathbf{f} = [f(\mathbf{x}_1), ..., f(\mathbf{x}_N)]^T$, has a Gaussian distribution $\mathbf{f} \sim \mathcal{N}(\boldsymbol{\mu}, \boldsymbol{K})$ with means $\boldsymbol{\mu}_i = m(\mathbf{x}_i)$ and covariance $\boldsymbol{K}_{ij} = k(\mathbf{x}_i, \mathbf{x}_j)$. The mean $m(\mathbf{x})$ is often set to be the constant zero function, and the kernel $k(\mathbf{x}, \mathbf{x}')$ is a positive semi-definite function. The target values are assumed to be independent when conditioned on $\mathbf{f}$. For Gaussian process regression the likelihood is Gaussian, $p(y|f) = \mathcal{N}(f, \sigma^2)$. Therefore, the posterior $p(f|y, \mathbf{X})$ is also Gaussian, and both the marginal and the predictive distributions have known analytic expressions. This is one of the main motivations behind using GPs, as most other Bayesian models have intractable inference.

Unfortunately, for Gaussian process classification (GPC) the likelihood, $p(y|f)$, is not a Gaussian and the posterior does not admit a closed-form expression. One approach for applying GPs to binary classification tasks is the Pólya-Gamma augmentation [57]. Using this approach, we can augment the GP model with random variables $\boldsymbol{\omega}$ from a Pólya-Gamma distribution, one for each example. As a result, $p(\mathbf{f}|\mathbf{y}, \mathbf{X}, \boldsymbol{\omega})$ is a Gaussian density and $p(\boldsymbol{\omega}|\mathbf{y}, \mathbf{X}, \mathbf{f})$ is a Pólya-Gamma density. This allows to use Gibbs sampling to efficiently sample from the posterior $p(\mathbf{f}, \boldsymbol{\omega}|\mathbf{y}, \mathbf{X})$ for inference and prediction. A key advantage of the Pólya-Gamma augmentation is that it benefits from fast mixing and has the ability of even a single value of $\boldsymbol{\omega}$ to capture much of the volume of the marginal distribution over function values [46]. Full equations and further details on the Pólya-Gamma augmentation scheme are given in Appendix A.1.

**Deep kernel learning (DKL).** The quality of the GP model heavily depends on the kernel function $k(\mathbf{x}_i, \mathbf{x}_j)$. For many data modalities, such as images, common kernels are not a good measure of semantic similarity. Therefore, in [10, 80] standard kernels are used over features outputted

by a neural network $g_\theta$. For example, the RBF kernel $k_\theta(\mathbf{x}_i, \mathbf{x}_j) = \exp\left(-\frac{||g_\theta(\mathbf{x}_i) - g_\theta(\mathbf{x}_j)||^2}{2\ell^2}\right)$. In regression, it is possible to directly backpropagate through the GP inference as it is given in closed-form. In our case, we use Fisher's identity [19] to obtain stochastic gradients [69].

**Inducing points.** GPs require storing and inverting a kernel matrix on the entire training set which often limits its usage. A common solution to this problem is to use inducing point methods [58, 70]. The key idea is to replace the exact kernel with an approximation for fast computation. Usually, $M \ll N$ pseudo-inputs are learned such that the main computational bottleneck is in inverting $M \times M$ matrices.

**GP-Tree.** We build on GP-Tree [2], a recent GP classifier that was shown to scale well with dataset size and the number of classes. GP-Tree turns the multi-class classification problem into a sequence of binary decisions along the tree nodes. Each node in the tree fits a binary GP classifier based on the Pólya-Gamma augmentation scheme and the data associated with that node. The leaf nodes correspond to the classes in the dataset. The tree is constructed by first computing a prototype for each class and then recursively performing divisive hierarchical clustering on these prototypes to two clusters at each node. Further details are given in Appendix A.2.

# 4 pFedGP: federated learning with Gaussian processes

Now we describe our approach for applying personalized federated learning (PFL) with Gaussian processes. First, we extend GP-Tree to the FL setup and show how to use Gibbs sampling to learn the NN parameters. Then, we present two alternatives for this method that use inducing points. The first is for extremely limited-size datasets, while the second allows controlling the computational resources. We name our method *pFedGP*. An illustration of our method is given in Figure 1.

## 4.1 A full GP model

The training procedure follows the standard protocol in this field [4, 44, 51]. We assume the existence of a server that holds the shared parameters $\theta$ (a NN). Let $C$ denote the set of clients. For each client $c \in C$ we denote by $D_c$ its local dataset of size $N_c$. At each training iteration (round) the model is sent to $S$ clients to perform kernel learning ($|S| \leq |C|$). Each client $c \in S$ updates its copy of the global model and then sends the updated model to the server. The server then averages over the updates to obtain a new global model.

At each client $c$, we perform kernel learning in the following manner. We first compute the feature representation of the data samples associated with the client using the shared network. Then, we build the hierarchical classification tree as discussed in Section 3 & Appendix A.2. In [2] the tree was built only once after a pre-training stage and the model parameters were learned using a variational inference approach. Here, we re-build the tree at each round using the most recent features and we use a Gibbs sampling procedure, as it allows this flexibility in building the tree and performs better when not prohibitive by computational limitations. Learning the network parameters $\theta$ with the Gibbs sampling approach can be done with two common objectives, the marginal likelihood, and the predictive distribution.

We denote by $\mathbf{X}_v$ the data associated with the tree node $v$, i.e., the data points which have $v$ on the path from the root node to their class leaf node. We denote by $\mathbf{y}_v$ the binary label of these points, i.e., does their path go left or right at this node. And we denote by $\boldsymbol{\omega}_v$ the Pólya-Gamma random variables associated with node $v$. The marginal likelihood term for the full hierarchical classification tree is the sum of the separate marginal likelihood terms of all the nodes $v$ in the tree:

$$\mathcal{L}_c^{ML}(\theta; D_c) = \sum_v \log p_\theta(\mathbf{y}_v | \mathbf{X}_v) = \sum_v \log \int p_\theta(\mathbf{y}_v | \boldsymbol{\omega}_v, \mathbf{X}_v) p(\boldsymbol{\omega}_v) d\boldsymbol{\omega}_v. \tag{1}$$

Similar to [69] we use a gradient estimator based on Fisher's identity [19]:

$$\nabla_\theta \mathcal{L}_c^{ML}(\theta; D_c) = \sum_v \int p_\theta(\boldsymbol{\omega}_v | \mathbf{y}_v, \mathbf{X}_v) \nabla_\theta \log p_\theta(\mathbf{y}_v | \boldsymbol{\omega}_v, \mathbf{X}_v) d\boldsymbol{\omega}_v \approx \sum_v \frac{1}{L} \sum_{l=1}^L \nabla_\theta \log p_\theta(\mathbf{y}_v | \boldsymbol{\omega}_v^{(l)}, \mathbf{X}_v). \tag{2}$$

Here, $\boldsymbol{\omega}_v^{(1)}, ..., \boldsymbol{\omega}_v^{(L)}$ are samples from the posterior at node $v$. Due to the Pólya-Gamma augmentation $p_\theta(\mathbf{y}_v | \boldsymbol{\omega}_v^{(l)}, \mathbf{X}_v)$ is proportional to a Gaussian density. The exact expression is give in Appendix A.2.

To use the predictive distribution as an objective, in each training iteration, after building the tree model, at each node we randomly draw a portion from the (node) training data and use it to predict the class label for the remaining part. We denote with $\mathbf{X}_v$ and $\mathbf{y}_v$ the training portion, $\mathbf{x}_v^*$ and $y_v^*$ the input and the label of the point we are predicting, and $P^{y^*}$ the path from the root node to the $y^*$ leaf node (i.e., the original class). Here we also take advantage of the independence between nodes to maximize the predictive distribution per node individually. The predictive distribution for a single data point:

$$\mathcal{L}_c^{PD}(\theta; \mathbf{x}^*, y^*) = \sum_{v \in P^{y^*}} \log\ p_\theta(y_v^* | \mathbf{x}_v^*, \mathbf{y}_v, \mathbf{X}_v) = \sum_{v \in P^{y^*}} \log\ \int p_\theta(y_v^* | \boldsymbol{\omega}_v, \mathbf{x}_v^*, \mathbf{y}_v, \mathbf{X}_v) p(\boldsymbol{\omega}_v | \mathbf{y}_v, \mathbf{X}_v) d\boldsymbol{\omega}_v.$$
(3)

We use an approximate-gradient estimator based on posterior samples of $\boldsymbol{\omega}$:

$$\nabla_\theta \mathcal{L}_c^{PD}(\theta; \mathbf{x}^*, y^*) \approx \sum_{v \in P^{y^*}} \frac{1}{L} \sum_{l=1}^{L} \nabla_\theta log\ p_\theta(y_v^* | \boldsymbol{\omega}_v^{(l)}, \mathbf{x}_v^*, \mathbf{y}_v, \mathbf{X}_v).$$
(4)

Where $p_\theta(y_v^* | \boldsymbol{\omega}_v^{(l)}, \mathbf{x}_v^*, \mathbf{y}_v, \mathbf{X}_v) = \int p(y_v^* | f^*) p_\theta(f^* | \boldsymbol{\omega}_v^{(l)}, \mathbf{x}_v^*, \mathbf{y}_v, \mathbf{X}_v) df^*$ does not have an analytical expression, but $p_\theta(f^* | \boldsymbol{\omega}_v^{(l)}, \mathbf{x}_v^*, \mathbf{y}_v, \mathbf{X}_v) = \int p_\theta(f^* | \mathbf{f}, \mathbf{x}_v^*, \mathbf{X}_v) p_\theta(\mathbf{f} | \boldsymbol{\omega}_v^{(l)}, \mathbf{y}_v, \mathbf{X}_v) df^*$ is Gaussian with known parameters. We then compute the predictive distribution by performing Gauss-Hermite integration over $f^*$. See exact expression in Appendix A.2.

## 4.2 Augmenting the model with inducing points: sample efficiency

The GP model described in Section 4.1 works well in most situations. However, when the number of data points per client is small, performance naturally degrades. To increase information sharing between clients and improve the per-client performance, we suggest augmenting the model with global inducing points shared across clients. When sending the model from the server to a client, we also send the inducing inputs and their labels. To streamline optimization and reduce the communication burden, we define the inducing inputs in the feature space of the last embedding layer of the shared NN. Therefore, usually, their size will be negligible compared to the network size.

We denote by $\bar{\mathbf{X}}$ the learned inducing inputs and by $\bar{\mathbf{y}}$ their fixed class labels. They are set evenly across classes. During training, we regard *only* the set of inducing inputs-labels $(\bar{\mathbf{X}}, \bar{\mathbf{y}})$ as the available (training) data and use them for posterior inference. More formally, we first compute $p_\theta(\mathbf{f} | \bar{\boldsymbol{\omega}}, \bar{\mathbf{y}}, \bar{\mathbf{X}}, \mathbf{X}) = \int p_\theta(\mathbf{f} | \bar{\mathbf{f}}, \bar{\mathbf{X}}, \mathbf{X}) p_\theta(\bar{\mathbf{f}} | \bar{\boldsymbol{\omega}}, \bar{\mathbf{y}}, \bar{\mathbf{X}}) d\bar{\mathbf{f}}$ using its analytical expression for the actual training data and then compute the probability of $\mathbf{y}$ using Gauss-Hermite integration. Then we use Eq. 3 & 4 for learning the network parameters and the inducing locations. At test time, to make full use of the training data, we combine the inducing inputs with the training data and use both to obtain the GP formulas and to make predictions. We note that with just using the inducing inputs at test time the model performs remarkably well, despite having almost no personalization component. See Appendix E.8 for a further discussion.

One potential issue with using IPs in this manner is that it distorts the true class distribution. As a result, the classifier may be more likely to predict a low-probability class during test time. We address this issue by adjusting the output distribution. In general, let $p(y, \mathbf{x})$ and $q(y, \mathbf{x})$ be two distributions that differ only in the class probabilities, i.e. $p(\mathbf{x}|y) = q(\mathbf{x}|y)$, the predictive distribution follows:

$$\frac{q(y^*|\mathbf{x}^*)}{p(y^*|\mathbf{x}^*)} \propto \frac{q(\mathbf{x}^*|y^*)q(y^*)}{p(\mathbf{x}^*|y^*)p(y^*)} \implies q(y^*|\mathbf{x}^*) \propto \frac{q(y^*)}{p(y^*)} p(y^*|\mathbf{x}^*).$$
(5)

We use this to correct the GP predictions to the original class ratios at each tree node. We found in our experiments that this correction generally improves the classifier performance for class imbalanced data. As an example for this phenomena, consider a binary classification problem having 90 examples from the first class and 10 examples from the second class (therefore, $q(y = 0) = 0.9$, and $q(y = 1) = 0.1$). Assume we defined 50 inducing inputs per class, so now during test time the model sees 140 samples from the first class and 60 samples from the second class which corresponds to probabilities $p(y = 0) = 0.7$ and $p(y = 1) = 0.3$.

## 4.3 Augmenting the model with inducing points: computational efficiency

Learning the full GP model described in Section 4.1 requires inverting a matrix of size $N_c$ in the worst case (at the root node), which has $\mathcal{O}(N_c^3)$ run-time complexity and $\mathcal{O}(N_c^2)$ memory complexity.

---

**Algorithm 1** *pFedGP*. $C$ clients indexed by c; $E$ - number of local epochs; $|S|$ - number of sampled clients; $M$ - number of inducing inputs per class

---

**Server executes:**
    Initialize shared network $\theta \leftarrow \theta_0$
    Initialize $M$ inducing inputs per class for all classes in the system # in pFedGP-IP variants only
    **for** each round $t \leftarrow 1, 2, ...$ **do**:
        Sample $S$ clients uniformly at random
        **for** each client $c \in S$ **in parallel**:
            $\theta_{t+1}^c, M_{t+1}^c \leftarrow ClientUpdate(\theta_t, M_t)$ # obtain updates from client c
        Update $\theta_{t+1}, M_{t+1}$ using FedAvg [51] update rule.

**ClientUpdate($\theta, M$):**
    **for** each local epoch $e \leftarrow 1, ..., E$ **do**:
        **if** $e = 1$:
            Build GP tree classifier using the personal dataset $D_c$
        Update $\theta, M$ using gradient-based optimization methods on $D_c$ with $\mathcal{L}^{ML}$ or $\mathcal{L}^{PD}$
    **return** $\theta, M$

---

Therefore, we propose an additional procedure based on inducing points to allow reduced complexity in low resource environments and scalability to larger dataset sizes.

This variant is based on the fully independent training conditional (FITC) method [70]. The key idea is to cast all the dependence on the inducing points and assume independence between the latent function values given the inducing points. Here for brevity, we omit the subscripts denoting the client and the tree node. However, all quantities and data points are those that belong to a specific client and tree node. Let $\bar{\mathbf{X}} \in \mathbb{R}^{M \times d}$ denote the pseudo-inputs (defined in the embedding space of the last layer of the NN), and $\bar{\mathbf{f}} \in \mathbb{R}^M$ the corresponding latent function values. Here as well, the inducing inputs are defined globally at the server level and they are set evenly across classes.

We assume the following GP prior $p(\mathbf{f}, \bar{\mathbf{f}}) = \mathcal{N}\left(\mathbf{0}, \begin{bmatrix} \mathbf{K}_{NN} & \mathbf{K}_{NM} \\ \mathbf{K}_{MN} & \mathbf{K}_{MM} \end{bmatrix}\right)$, where $\mathbf{K}_{MM}$ is the kernel between the inducing inputs, $\mathbf{K}_{NN}$ is a diagonal matrix between the actual training data, $\mathbf{K}_{NM}$ is the kernel between the data and the inducing inputs, and we placed a zero mean prior. The likelihood of the dataset when factoring the inducing variables and the Pólya-Gamma variables (one per training sample), and the posterior over $\mathbf{f}$, both have known analytical expressions. We can then obtain the posterior and marginal distributions by marginalizing over $\bar{\mathbf{f}}$. Here we will present the posterior and marginal distributions:

$$p(\mathbf{f}|\mathbf{X}, \mathbf{y}, \boldsymbol{\omega}, \bar{\mathbf{X}}) = \mathcal{N}(\mathbf{f}|\mathbf{K}_{NM}\mathbf{B}^{-1}\mathbf{K}_{MN}\boldsymbol{\Lambda}^{-1}\boldsymbol{\Omega}^{-1}\boldsymbol{\kappa}, \ \mathbf{K}_{NN} - \mathbf{K}_{NM}(\mathbf{K}_{MM}^{-1} - \mathbf{B}^{-1})\mathbf{K}_{MN}), \tag{6}$$

$$p(\mathbf{y}|\mathbf{X}, \boldsymbol{\omega}, \bar{\mathbf{X}}) \propto \mathcal{N}(\boldsymbol{\Omega}^{-1}\boldsymbol{\kappa}|\mathbf{0}, \ \boldsymbol{\Lambda} + \mathbf{K}_{NM}\mathbf{K}_{MM}^{-1}\mathbf{K}_{MN}). \tag{7}$$

Where $\boldsymbol{\Omega} = diag(\boldsymbol{\omega})$, $\kappa_j = y_j - 1/2$, $\boldsymbol{\Lambda} = \boldsymbol{\Omega}^{-1} + diag(\mathbf{K}_{NN} - \mathbf{K}_{NM}\mathbf{K}_{MM}^{-1}\mathbf{K}_{MN})$, and $\mathbf{B} = \mathbf{K}_{MM} + \mathbf{K}_{MN}\boldsymbol{\Lambda}^{-1}\mathbf{K}_{NM}$. Importantly, we only need to invert $M \times M$ or diagonal matrices. See full derivation in Appendix B. During test time, we use $\bar{\mathbf{f}}$ to get the posterior of $f^*$ to compute the predictive distribution.

Now we can use either the marginal or the predictive distribution to learn the shared NN parameters and the inducing locations. The complexity of applying this procedure is reduced to $\mathcal{O}(M^2 N_c + M^3)$ in run-time, and $\mathcal{O}(M N_c + M^2)$ in memory. While the (conditional) independence assumption between the latent function values may be restrictive, we found this method to be comparable with the full GP alternative in our experiments. Potentially, this can be attributed to the effect of sharing the inducing inputs among clients and the information that $\boldsymbol{\omega}$ stores on $\mathbf{f}$.

## 5 Generalization bound

It is reasonable to expect that after we learned the system new clients will arrive. In such cases, we would like to use pFedGP without re-training the kernel function. Under this scenario, we can derive generalization bounds concerning only the GP classifier without taking into account the fixed

neural network using PAC-Bayes bound [50]. Having meaningful guarantees can be very important in safety-critical applications. The PAC-Bayes bound for GPC [64] (with the Gibbs risk):

**Theorem 1.** *Given i.i.d. samples $D_c = \{(\mathbf{x}_i, y_i)\}_{i=1}^{N_c}$ of size $N_c$ drawn from any data distribution over $\mathcal{X} \times \{-1, 1\}$, a GP posterior Q, and a GP prior P, the following bound holds, where the probability is over random data samples:*

$$Pr_{D_c}\{R(Q) > R_{D_c}(Q) + KL_{ber}^{-1}(R_{D_c}(Q), \epsilon(\delta, n, P, Q))\} \le \delta. \qquad (8)$$

*Here, we have,*

$$R(Q) = \mathbb{E}_{(\mathbf{x}^*, y^*)}[Pr_{f^* \sim Q(f^*|\mathbf{x}^*, D_c)}\{sign\, f^* \ne y^*\}], \quad R_{D_c}(Q) = \frac{1}{N_c}\sum_{i=1}^{N_c} Pr_{f_i \sim Q(f_i|D_c)}\{sign\, f_i \ne y_i\}$$

$$\epsilon(\delta, N_c, P, Q) = \frac{1}{N_c}\Big(KL[Q \,||\, P] + \log \frac{N_c + 1}{\delta}\Big), \quad KL_{ber}^{-1}(q, \epsilon) = max_{p \in [0,1]} KL_{ber}[q \,||\, p] \le \epsilon$$

$$(9)$$

An important observation in [64] is that the KL-divergence between the posterior and prior Gaussian processes is equivalent to the KL-divergence between the posterior and prior distribution of their values on the $N_c$ training samples. While [64] assumed $Q$ to be Gaussian, this observation still holds even without this assumption. However, when $Q$ is no longer Gaussian, as is the case here, $KL[Q(\mathbf{f}) \,||\, P(\mathbf{f})]$ no longer has a closed-form expression. We can show that for the Pólya-Gamma augmentation:

$$KL[Q(\mathbf{f}) \,||\, P(\mathbf{f})] = \mathbb{E}_{Q(\boldsymbol{\omega})}\{KL[Q(\mathbf{f}|\boldsymbol{\omega})||P(\mathbf{f})]\} - MI[\mathbf{f}; \boldsymbol{\omega}]$$

$$= \mathbb{E}_{Q(\boldsymbol{\omega})}\{KL[Q(\mathbf{f}|\boldsymbol{\omega})||P(\mathbf{f})]\} + \mathbb{E}_{Q(\mathbf{f}, \boldsymbol{\omega})}\left[\log \frac{Q(\boldsymbol{\omega})}{Q(\boldsymbol{\omega}|\mathbf{f})}\right] \qquad (10)$$

where MI denotes the mutual information. Since $Q(\mathbf{f}|\boldsymbol{\omega})$ and $P(\mathbf{f})$ are Gaussian, the $KL[Q(\mathbf{f}|\boldsymbol{\omega})||P(\mathbf{f})]$ term has a close form expression so we only need to perform Monte-Carlo approximation on the expectation on $\boldsymbol{\omega}$ on the first element. In the second expectation, $Q(\boldsymbol{\omega})$ does not have a known expression. To estimate it, given $\{(\boldsymbol{\omega}_i, \mathbf{f}_i)\}_{i=1}^N$ samples, we use $Q(\boldsymbol{\omega}_i) \approx \frac{1}{N-1}\sum_{j \ne i} Q(\boldsymbol{\omega}_i|\mathbf{f}_j)$. Note that if the summation for $j$ includes $\mathbf{f}_i$, it might result in a biased estimator. Further details on estimating $KL[Q(\mathbf{f}) \,||\, P(\mathbf{f})]$ are in Appendix C.

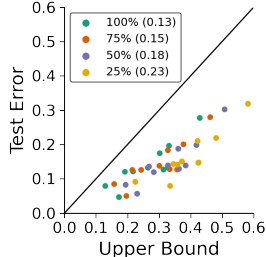

Figure 2: Test error vs an estimated upper bound over 10 clients with varying degrees of a training set data size. Each dot represents a combination of client and data size. In parenthesis - the average difference between the empirical and the test error.

To assess the quality of the bound, we partitioned the CIFAR-10 dataset to 100 clients. We trained a shared network using our full-GP variant on 90 clients and then recorded the generalization and test error on the remaining 10 clients four times, each with a different training set size. Figure 2 shows the estimation of the generalization error bound ($\delta = 0.01$) vs the actual error on the novel clients with the Gibbs classifier. First, we observe that indeed the bound is greater than the actual test error for all points and that it is not vacuous. There is a strong correlation between the actual error and the bound. Secondly, unlike worst-case bounds (e.g., VC-dimension), this bound depends on the actual data and not only the number of data points.

## 6 Experiments

We evaluated pFedGP against baseline methods in various learning setups. We present the result for the following model variants: *(i) pFedGP*, the full GP model (Section 4.1); *(ii) pFedGP-IP-data*, the model with IPs described in Section 4.2; and *(iii) pFedGP-IP-compute*, the model with IPs described in Section 4.3. For pFedGP and pFedGP-IP-compute, the results obtained by maximizing the predictive and marginal likelihood were similar, with a slight advantage to the former. Therefore, we present here the results only for the predictive alternative and defer the results of the marginal alternative to the Appendix. Additional experiments, ablation study, and further analyses are provided

Table 1: Test accuracy ($\pm$ SEM) over 50, 100, 500 clients on CIFAR-10, CIFAR-100, and CINIC-10. The *# samples/client* indicates the average number of training samples per client.

| | CIFAR-10 | | | CIFAR-100 | | | CINIC-10 | | |
|---|---|---|---|---|---|---|---|---|---|
| # clients | 50 | 100 | 500 | 50 | 100 | 500 | 50 | 100 | 500 |
| # samples/client | 800 | 400 | 80 | 800 | 400 | 80 | 1800 | 900 | 180 |
| Local | $86.2 \pm 0.2$ | $82.9 \pm 0.4$ | $74.8 \pm 0.5$ | $52.1 \pm 0.2$ | $45.6 \pm 0.3$ | $30.9 \pm 0.2$ | $61.1 \pm 0.3$ | $56.9 \pm 0.7$ | $46.4 \pm 0.1$ |
| FedAvg [51] | $56.4 \pm 0.5$ | $59.7 \pm 0.5$ | $54.0 \pm 0.5$ | $23.6 \pm 0.2$ | $24.0 \pm 0.2$ | $20.4 \pm 0.0$ | $45.6 \pm 0.4$ | $44.7 \pm 0.5$ | $45.7 \pm 0.5$ |
| FOLA [47] | $55.9 \pm 3.3$ | $52.1 \pm 3.1$ | $45.9 \pm 0.3$ | $25.5 \pm 1.5$ | $22.4 \pm 1.3$ | $18.7 \pm 0.1$ | $45.2 \pm 0.3$ | $43.4 \pm 0.3$ | $38.3 \pm 0.2$ |
| FedPer [4] | $83.8 \pm 0.8$ | $81.5 \pm 0.5$ | $76.8 \pm 1.2$ | $48.3 \pm 0.6$ | $43.6 \pm 0.2$ | $25.6 \pm 0.3$ | $70.6 \pm 0.2$ | $68.4 \pm 0.5$ | $62.2 \pm .05$ |
| LG-FedAvg [44] | $87.9 \pm 0.3$ | $83.6 \pm 0.7$ | $64.7 \pm 0.7$ | $43.6 \pm 0.2$ | $37.5 \pm 0.9$ | $20.3 \pm 0.5$ | $59.5 \pm 1.1$ | $59.9 \pm 2.1$ | $52.5 \pm 0.8$ |
| pFedMe [72] | $86.4 \pm 0.8$ | $85.0 \pm 0.3$ | $80.3 \pm 0.5$ | $49.8 \pm 0.5$ | $47.7 \pm 0.4$ | $32.5 \pm 0.8$ | $69.9 \pm 0.5$ | $68.9 \pm 0.7$ | $58.8 \pm 0.1$ |
| FedU [18] | $80.6 \pm 0.3$ | $78.1 \pm 0.5$ | $65.6 \pm 0.4$ | $41.1 \pm 0.2$ | $36.0 \pm 0.2$ | $15.9 \pm 0.4$ | $59.3 \pm 0.2$ | $55.4 \pm 0.6$ | $41.6 \pm 0.5$ |
| pFedHN [65] | $\mathbf{90.2 \pm 0.6}$ | $87.4 \pm 0.2$ | $83.2 \pm 0.8$ | $60.0 \pm 1.0$ | $52.3 \pm 0.5$ | $34.1 \pm 0.1$ | $70.4 \pm 0.4$ | $69.4 \pm 0.5$ | $64.2 \pm .05$ |
| **Ours** | | | | | | | | | |
| pFedGP-IP-data | $88.6 \pm 0.2$ | $87.4 \pm 0.2$ | $86.9 \pm 0.7$ | $60.2 \pm 0.3$ | $58.5 \pm 0.3$ | $\mathbf{55.7 \pm 0.4}$ | $69.8 \pm 0.2$ | $68.3 \pm 0.6$ | $67.6 \pm 0.3$ |
| pFedGP-IP-compute | $89.9 \pm 0.6$ | $\mathbf{88.8 \pm 0.1}$ | $86.8 \pm 0.4$ | $61.2 \pm 0.4$ | $59.8 \pm 0.3$ | $49.2 \pm 0.3$ | $\mathbf{72.0 \pm 0.3}$ | $\mathbf{71.5 \pm 0.5}$ | $\mathbf{68.2 \pm 0.2}$ |
| pFedGP | $89.2 \pm 0.3$ | $\mathbf{88.8 \pm 0.2}$ | $\mathbf{87.6 \pm 0.4}$ | $\mathbf{63.3 \pm 0.1}$ | $\mathbf{61.3 \pm 0.2}$ | $50.6 \pm 0.2$ | $\mathbf{71.8 \pm 0.3}$ | $71.3 \pm 0.4$ | $68.1 \pm 0.3$ |

in Appendix E. Unless stated otherwise, we report the average and the standard error of the mean (SEM) over three random seeds of the federated accuracy, defined as the average accuracy across all clients and samples.

**Datasets.** All methods were evaluated on CIFAR-10, CIFAR-100 [38], and CINIC-10 [15] datasets. CINIC-10 is more diverse since it combines images from CIFAR-10 and ImageNet [16].

**Compared methods.** We compared our method against the following baselines: *(1) Local*, pFedGP full model on each client with a private network and no collaboration with other clients; *(2) FedAvg* [51], a standard FL model with no personalization component; *(3) FOLA* [47], a Bayesian method that used a multivariate Gaussian product mechanism to aggregate local models; *(4) FedPer* [4], a PFL approach that learns a personal classifier for each client on top of a shared feature extractor; *(5) LG-FedAvg* [44], a PFL method that uses local feature extractor per client and global output layers; *(6) pFedMe* [72], a PFL method which adds a Moreau-envelopes loss term; *(7) FedU* [18], a recent multi-task learning approach for PFL that learns a model per client; *(8) pFedHN* [65], a recent PFL approach that uses a hypernetwork to generate client-specific networks.

**Training protocol.** We follow the training strategy proposed in [65]. We limit the training process to 1000 communication rounds, in each we sample five clients uniformly at random for model updates. The training procedure is different in the FOLA and pFedHN baselines, so we used an equivalent communication cost. In LG-FedAvg, we made an extra 200 communication rounds after a pre-training stage with the FedAvg model for 1000 communication rounds. In the local model, we performed 100 epochs of training for each client. In all experiments, we used a LeNet-based network [40] having two convolution layers followed by two fully connected layers and an additional linear layer. We tuned the hyperparameters of all methods using a pre-allocated held-out validation set. Full experimental details are given in Appendix D.

### 6.1 Standard PFL setting

We first evaluated all methods in a standard PFL setting [65, 72]. We varied the total number of clients in the system from 50 to 500 and we set the number of classes per client to two/ten/four for CIFAR-10/CIFAR-100/CINIC-10 respectively. Since the total number of samples in the system is fixed, the number of samples per client changed accordingly. For each client, the same classes appeared in the training and test set.

The results are presented in Table 1. They show that: (1) The performance of the *local* baseline is significantly impaired when the number of samples per client decreases, emphasizing the importance of federated learning in the presence of limited local data. (2) FedAvg and FOLA, which do not use personalized FL, perform poorly in this heterogeneous setup. (3) pFedGP outperforms or is on par with previous state-of-the-art approaches when local data is sufficient (e.g., 50 clients on all datasets). When the data per client becomes limited, pFedGP achieves significant improvements over competing methods; note the $9\%$ and $21\%$ difference in CIFAR-100 over 100 and 500 clients, respectively. (4) pFedGP-IP-compute often achieves comparable results to pFedGP and is often superior to pFedGP-IP-data. We believe that it can be attributed to the fact that in pFedGP-IP-compute

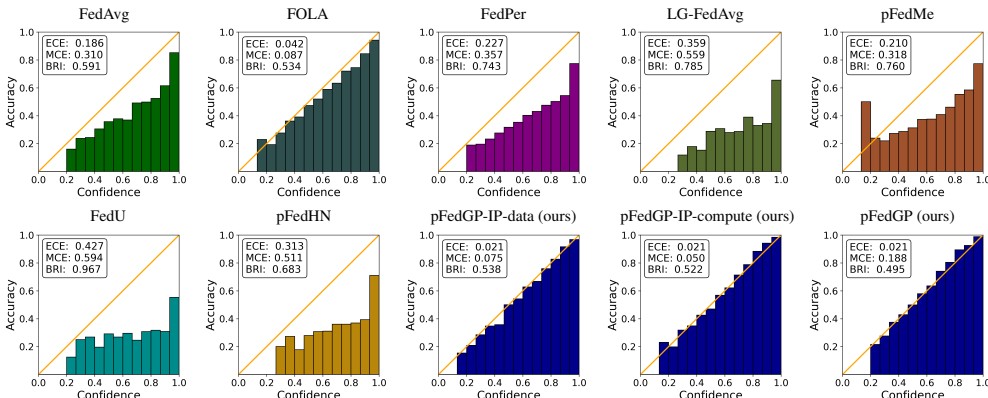

Figure 3: Reliability diagrams on CIFAR-100 with 50 clients. Diagonal indicates perfect calibration. Each plot also shows the expected & maximum calibration error (ECE & MCE) and the Brier Score (BRI). Lower is better.

the training data take an active part in the GP inference formulas (Eq. 6), while in pFedGP-IP-data the data impact in a weak manner only through the loss function. (5) pFedGP-IP-data is especially helpful when few samples per class are available, e.g., CIFAR-100 with 500 clients. That last point is further illustrated by decoupling the effect of the *number of clients* from that of the *training set size*. To illustrate that, in Appendix E.2 we fixed the number of clients and varied the number of training samples per class. From this experiment, we deduced that both factors (individually) contribute to pFedGP success.

A desired property from PFL classifiers is the ability to provide uncertainty estimation. For example, in decision support systems, such as in healthcare applications, the decision-maker should have an accurate estimation of the classifier confidence in the prediction. Here, we quantify the uncertainty through calibration. Figure 3 compares all methods both visually and using common metrics [7, 25, 55] on the CIFAR-100 dataset with 50 clients. Expected calibration error (ECE) measures the weighted average between the classifier confidence and accuracy. Maximum calibration error (MCE) takes the maximum instead of the average. And, Brier score (BRI) [7] measures the average squared error between the labels and the prediction probabilities. The figure shows that pFedGP classifiers are best calibrated across all metrics in almost all cases. We note that with temperature scaling, the calibration of the baseline methods can be improved [25]; however, choosing the right temperature requires optimization over a separate validation set, which our model does not need. Additional calibration results, including temperature scaling, are presented in Appendix E.9.

Table 2: Test accuracy ($\pm$ SEM) over 100 clients on noisy CIFAR-100. We also provide the relative accuracy decrease (%) w.r.t. the performance on the original CIFAR-100 data (see Table 1).

| Method | Accuracy | Decrease (%) |
| --- | --- | --- |
| FedPer [4] | $28.1 \pm 0.9$ | -35.6 |
| LG-FedAvg [44] | $26.9 \pm 0.9$ | -28.3 |
| pFedme [72] | $33.2 \pm 0.6$ | -30.4 |
| FedU [18] | $35.0 \pm 0.2$ | **-2.8** |
| pFedHN [65] | $38.9 \pm 0.5$ | -25.7 |
| **Ours** | | |
| pFedGP-IP-data | $45.0 \pm 0.3$ | -23.1 |
| pFedGP-IP-compute | $47.1 \pm .05$ | -21.2 |
| pFedGP | **$49.5 \pm 0.1$** | -19.2 |

### 6.2 PFL with input noise

In real-world federated systems, the clients may employ different measurement devices for data collection (cameras, sensors, etc.), resulting in different input noise characteristics per client. Here, we investigate pFedGP performance in this type of personalization. To simulate that, we partitioned CIFAR-10/100 to 100 clients similar to the protocol described in Section 6.1, we defined 57 unique distributions of image corruption noise [29], and we assigned a noise model to each client. Then for each example in each client, we sampled a corruption noise according to the noise model allocated to that client. Here we show the results for the noisy CIFAR-100 dataset in Table 2. Further details

on the perturbations performed and result for the noisy CIFAR-10 are given in the Appendix[2]. We observe a significant gap in favor of the pFedGP variants compared to baseline methods. Note that using global inducing points is slightly less beneficial in this case since they are defined globally and therefore are not tied to a specific noise type as the real client data is.

### 6.3  Generalization to out-of-distribution (OOD) novel clients

FL are dynamic systems. For example, novel clients may enter the system after the model was trained, possibly with a data distribution shift. Adapting to a new OOD client is both challenging and important for real-world FL systems. To evaluate pFedGP in this scenario, we followed the learning protocol proposed in [65]. We partitioned the CIFAR-10 dataset into two groups. The data in the first group was distributed between 90 clients for model training. The remaining data from the second group was distributed between an additional 10 clients that were excluded during training. Within each group, we set the class probabilities in each client by sampling from a Dirichlet distribution with the same $\alpha$ parameter. For the training group, we set $\alpha = 0.1$, trained the shared model using these clients, and froze it. Then, we evaluated the models on the second group by varying $\alpha \in \{.1, .25, .5, .75, 1\}$, on the remaining 10 clients. As $\alpha$ moves away from 0.1 the distribution shift between the two groups increases, resulting in more challenging OOD clients. Figure 4 reports the generalization gap as a function of the Dirichlet parameter $\alpha$. The generalization gap is computed by taking the difference between the average test accuracy of the (ten) novel clients and the average test accuracy of the (ninety) clients used for training. From the figure, here as well, pFedGP achieves the best generalization performance for all values of $\alpha$. Moreover, unlike baseline methods, pFedGP does not require *any* parameter tuning. Several baselines were excluded from the figure since they had a large generalization gap.

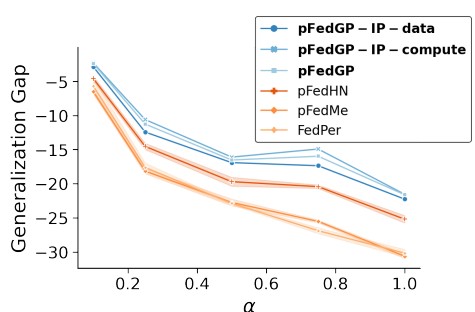

Figure 4:  Generalization to novel clients on CIFAR-10.

## 7  Conclusion

In this study, we proposed pFedGP, a novel method for PFL. pFedGP learns a kernel function, parameterized by a NN, that is shared between all clients using a personal GP classifier on each client. We proposed three variants for pFedGP, a full model approach that generally shows the best performance and two extensions to it. The first is most beneficial when the number of examples per class are small while the second allows controlling the computational requirements of the model. We also derived PAC-Bayes generalization bound on novel clients and empirically showed that it gives non-vacuous guarantees. pFedGP provides well-calibrated predictions, generalizes well to OOD novel clients, and consistently outperforms competing methods.

**Broader impact:** Our method shares the standard communication procedure of FL approaches, where no private data is directly communicated across the different nodes in the system. This protocol does not explicitly guarantee that no private information can be inferred at this time. As we show, pFedGP is particularly useful for clients with little data, and for clients that have strongly different distribution. This has great potential to improve client personalization in real-world systems, and do better at handling less common data. The latter is of great interest for decision support systems in sensitive domains such as health care or legal.

## Acknowledgements

This study was funded by a grant to GC from the Israel Science Foundation (ISF 737/2018), and by an equipment grant to GC and Bar-Ilan University from the Israel Science Foundation (ISF 2332/18). IA was funded by a grant from the Israeli innovation authority, through the AVATAR consortium.

---

[2]The noisy CIFAR-10/100 datasets are available at: https://idanachituve.github.io/projects/pFedGP

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
