# A Extended background

## A.1 The Pólya-Gamma augmentation

A random variable $\omega$ has a Pólya-Gamma distribution if it can be written as an infinite sum of independent gamma random variables:

$$\omega \overset{D}{=} \frac{1}{2\pi^2} \sum_{k=1}^{\infty} \frac{g_k}{(k-1/2)^2 + c^2/(4\pi^2)}, \tag{11}$$

where $b > 0$, $c \in \mathbb{R}$, and $g_k \sim Gamma(b,\ 1)$.

This random variable was proposed in [57] as it has the following desired property - For $b > 0$ the following identity holds:

$$\frac{(e^f)^a}{(1+e^f)^b} = 2^{-b} e^{\kappa f} \mathbb{E}_\omega[e^{-\omega f^2/2}], \tag{12}$$

where $\kappa = a - b/2$, and $\omega$ has the Pólya-Gamma distribution, $\omega \sim PG(b,\ 0)$. In [57] the authors also devised an efficient sampling algorithms for Pólya-Gamma random variables.

Suppose we are given with latent function values $\mathbf{f} \in \mathbb{R}^N$ having a binary classification assignment $\mathbf{y} \in \{0,1\}^N$. Let the prior over $\mathbf{f} \sim \mathcal{N}(\boldsymbol{\mu},\ \mathbf{K})$. The likelihood can be written as,

$$p(\mathbf{y}|\mathbf{f}) = \prod_{j=1}^{n} \sigma(f_j)^{y_j}(1-\sigma(f_j))^{1-y_j} = \prod_{j=1}^{n} \frac{(e^{f_j})^{y_j}}{1+e^{f_j}} = \mathbb{E}_{\boldsymbol{\omega}}\left[2^{-n}\exp\left(\sum_{j=1}^{n}\kappa_j f_j - \frac{\omega_j f_j^2}{2}\right)\right], \tag{13}$$

where $\sigma(\cdot)$ is the sigmoid function. Namely, we used Eq. 12 to augment the model with Pólya-Gamma variables (one per sample) such that the original likelihood is recovered when marginalizing them out.

Now, the augmented likelihood, $p(\mathbf{y}|\mathbf{f}, \boldsymbol{\omega})$, is proportional to a diagonal Gaussian and the posteriors in the augmented space have known expressions:

$$p(\mathbf{f}|\mathbf{y}, \boldsymbol{\omega}) = \mathcal{N}(\mathbf{f}|\boldsymbol{\Sigma}(\mathbf{K}^{-1}\boldsymbol{\mu} + \boldsymbol{\kappa}),\ \boldsymbol{\Sigma}),$$
$$p(\boldsymbol{\omega}|\mathbf{y}, \mathbf{f}) = PG(\mathbf{1},\ \mathbf{f}). \tag{14}$$

Where $\kappa_j = y_j - 1/2$, $\boldsymbol{\Sigma} = (\mathbf{K}^{-1} + \boldsymbol{\Omega})^{-1}$, and $\boldsymbol{\Omega} = diag(\boldsymbol{\omega})$. We can now sample from $p(\mathbf{f}, \boldsymbol{\omega}|\mathbf{y}, \mathbf{X})$ using block Gibbs sampling and get Monte-Carlo estimations of the marginal and predictive distributions.

## A.2 GP-Tree

Our method builds upon the method presented in [2]. It was shown to scale well both with dataset size and the number of classes, outperforming other GPC methods on standard benchmarks. We provide here a summary of this method, termed GP-Tree. GP-Tree uses the Pólya-Gamma augmentation, which is designed for binary classification tasks, in a (binary) tree-structure hierarchical model for multi-class classification tasks. Given a training dataset $D = (\mathbf{X}, \mathbf{y})$ of features and corresponding labels from $\{1, ..., \mathrm{T}\}$ classes, $D$ is partitioned recursively to two subsets, according to classes, at each tree level until reaching leaf nodes with data from only one class. More concretely, initially, feature vectors for all samples are obtained (using a NN), then a class prototype is generated by averaging the feature vectors belonging to the same class for all classes. Finally, a tree is formed using the divisive hierarchical clustering algorithm k-means++ [5] with $k = 2$ on the class prototypes. Partitioning the data in this manner is sensible since NNs tend to generate points that cluster around a single prototype for each class [68]. After building the tree, a GP model is assigned to each node to make a binary decision based on the data associated with that node. Let $f_v \sim \mathcal{GP}(m_v, k_v)$ be the GP associated with node $v$. We denote all the GPs in the tree with $\mathcal{F}$. The induced likelihood of a data point having the class $t$ is given by the unique path $P^t$ from the root to the leaf node corresponding to that class:

$$p(y = t|\mathcal{F}) = \prod_{v \in P^t} \sigma(f_v)^{y_v}(1-\sigma(f_v))^{1-y_v}, \tag{15}$$

where $y_v = 1$ if the path goes left at $v$ and zero otherwise. $y_v$ can be viewed as the (local) node label assignment of the example. Since this likelihood factorizes over the nodes, we can look at the nodes separately. Therefore, in the following, we will omit the subscript $v$ for brevity; however, all datum and quantities are those that belong to a specific node $v$.

In [2] two methods for applying GP inference were suggested: a variational inference (VI) approach and a Gibbs sampling procedure. The former is used when datasets are large by constructing a variational lower bound to learn the model parameters (e.g., the NN parameters), while the latter is used for Bayesian inference only when using a fixed model. Here we will focus on learning and inference with the Gibbs sampling procedure only (see main text for further details). To obtain the augmented marginal distribution and augmented predictive distribution for a novel point $\mathbf{x}^*$ at each node, we can sample $\boldsymbol{\omega}$ (a vector for each node) and use the following rules:

$$
\begin{aligned}
p(\mathbf{y}|\boldsymbol{\omega}, \mathbf{X}) &= \int p(\mathbf{y}|\boldsymbol{\omega}, \mathbf{X}, \mathbf{f})p(\mathbf{f})d\mathbf{f} \\
&\propto \mathcal{N}(\boldsymbol{\Omega}^{-1}\boldsymbol{\kappa}|\mathbf{0}, \mathbf{K} + \boldsymbol{\Omega}^{-1}),
\end{aligned}
\tag{16}
$$

$$
\begin{aligned}
p(f^*|\mathbf{x}^*, \mathbf{X}, \mathbf{y}, \boldsymbol{\omega}) &= \mathcal{N}(f^*|\mu^*, \Sigma^*), \\
\mu^* &= (\mathbf{k}^*)^T(\boldsymbol{\Omega}^{-1} + \mathbf{K})^{-1}\boldsymbol{\Omega}^{-1}\boldsymbol{\kappa}, \\
\Sigma^* &= k^{**} - (\mathbf{k}^*)^T(\boldsymbol{\Omega}^{-1} + \mathbf{K})^{-1}\mathbf{k}^*,
\end{aligned}
\tag{17}
$$

$$
p(y^*|\mathbf{x}^*, \mathbf{X}, \mathbf{y}, \boldsymbol{\omega}) = \int p(y^*|f^*)p(f^*|\mathbf{x}^*, \mathbf{X}, \mathbf{y}, \boldsymbol{\omega})df^*.
\tag{18}
$$

Where we assumed a zero mean prior, $k^{**} = k(\mathbf{x}^*, \mathbf{x}^*)$, $\mathbf{k}^*[i] = k(\mathbf{x}_i, \mathbf{x}^*)$, and $\mathbf{K}[i,j] = k(\mathbf{x}_i, \mathbf{x}_j)$. The integral in Eq. 18 is intractable, but can be computed numerically with 1D Gaussian-Hermite quadrature.

## B  pFedGP-IP-compute detailed derivation

Here we describe in more detail our pFedGP-IP-compute variant. The key idea behind this method is to cast all the dependence on the inducing points and assume independence between the latent function values given the inducing points. Since the inference problem factorizes over the clients and tree nodes, we may compute all quantities separately for each client and tree node and only afterward aggregate the results. Therefore, in the below, we omit the subscripts denoting the client and node; however, all quantities belong to a specific client and node. You may recall that $\bar{\mathbf{X}} \in \mathbb{R}^{M \times d}$ denote the pseudo-inputs, defined in the embedding space of the last layer of the NN and are shared by all clients, and $\bar{\mathbf{f}} \in \mathbb{R}^M$ are the corresponding latent function values. We assume the following GP prior $p(\mathbf{f}, \bar{\mathbf{f}}) = \mathcal{N}(\mathbf{0}, \begin{bmatrix} \mathbf{K}_{NN} & \mathbf{K}_{NM} \\ \mathbf{K}_{MN} & \mathbf{K}_{MM} \end{bmatrix})$. The data likelihood of the dataset when factoring the inducing variables and the Pólya-Gamma variables (one per training sample) is proportional to a Gaussian:

$$
p(\mathbf{y}|\mathbf{X}, \boldsymbol{\omega}, \bar{\mathbf{f}}, \bar{\mathbf{X}}) = \prod_{n=1}^{N} p(y_n|\mathbf{x}_n, \boldsymbol{\omega}, \bar{\mathbf{f}}, \bar{\mathbf{X}}) \propto \mathcal{N}(\boldsymbol{\Omega}^{-1}\boldsymbol{\kappa}|\mathbf{K}_{NM}\mathbf{K}_{MM}^{-1}\bar{\mathbf{f}}, \boldsymbol{\Lambda}),
\tag{19}
$$

where, $\boldsymbol{\Omega} = diag(\boldsymbol{\omega})$, $\kappa_j = y_j - 1/2$, and $\boldsymbol{\Lambda} = \boldsymbol{\Omega}^{-1} + diag(\mathbf{K}_{NN} - \mathbf{K}_{NM}\mathbf{K}_{MM}^{-1}\mathbf{K}_{MN})$.

The posterior over $\bar{\mathbf{f}}$ is obtained using Bayes rule:

$$
p(\bar{\mathbf{f}}|\boldsymbol{\omega}, \mathbf{y}, \mathbf{X}, \bar{\mathbf{X}}) = \mathcal{N}(\bar{\mathbf{f}}|\mathbf{K}_{MM}\mathbf{B}^{-1}\mathbf{K}_{MN}\boldsymbol{\Lambda}^{-1}\boldsymbol{\Omega}^{-1}\boldsymbol{\kappa}, \mathbf{K}_{MM}\mathbf{B}^{-1}\mathbf{K}_{MM}),
\tag{20}
$$

where $\mathbf{B} = \mathbf{K}_{MM} + \mathbf{K}_{MN}\boldsymbol{\Lambda}^{-1}\mathbf{K}_{NM}$.

The posterior distribution over $\mathbf{f}$ can be obtained by marginalization over $\bar{\mathbf{f}}$:

$$
\begin{aligned}
p(\mathbf{f}|\boldsymbol{\omega}, \mathbf{y}, \mathbf{X}, \bar{\mathbf{X}}) &= \int p(\mathbf{f}|\bar{\mathbf{f}}, \mathbf{X}, \bar{\mathbf{X}})p(\bar{\mathbf{f}}|\boldsymbol{\omega}, \mathbf{y}, \mathbf{X}, \bar{\mathbf{X}})d\bar{\mathbf{f}} \\
&= \mathcal{N}(\mathbf{f}|\mathbf{K}_{NM}\mathbf{B}^{-1}\mathbf{K}_{MN}\boldsymbol{\Lambda}^{-1}\boldsymbol{\Omega}^{-1}\boldsymbol{\kappa}, \mathbf{K}_{NN} - \mathbf{K}_{NM}(\mathbf{K}_{MM}^{-1} - \mathbf{B}^{-1})\mathbf{K}_{MN}).
\end{aligned}
\tag{21}
$$

We note that while we use $\bar{\mathbf{f}}$ for predictions, we still need $\mathbf{f}$ in order to sample $\boldsymbol{\omega}$.

Given $\boldsymbol{\omega}$ and the expression for the posterior over $\bar{\mathbf{f}}$, we can compute the predictive distribution for a novel input $\mathbf{x}^*$:

$$p(f^*|\mathbf{x}^*, \boldsymbol{\omega}, \mathbf{y}, \mathbf{X}, \bar{\mathbf{X}}) = \int p(f^*|\mathbf{x}^*, \bar{\mathbf{f}}) p(\bar{\mathbf{f}}|\boldsymbol{\omega}, \mathbf{y}, \mathbf{X}, \bar{\mathbf{X}}) d\bar{\mathbf{f}}$$

$$= \mathcal{N}(f^*|(\mathbf{k}^*)^T \mathbf{B}^{-1} \mathbf{K}_{MN} \boldsymbol{\Lambda}^{-1} \boldsymbol{\Omega}^{-1} \boldsymbol{\kappa}, \ k^{**} - (\mathbf{k}^*)^T (\mathbf{K}_{MM}^{-1} - \mathbf{B}^{-1}) \mathbf{k}^*), \tag{22}$$

where $k^{**} = k(\mathbf{x}^*, \mathbf{x}^*)$, and $\mathbf{k}^*[i] = k(\bar{\mathbf{x}}_i, \mathbf{x}^*)$.

The marginal distribution is given by:

$$p(\mathbf{y}|\boldsymbol{\omega}, \mathbf{X}, \bar{\mathbf{X}}) = \int p(\mathbf{y}|\boldsymbol{\omega}, \bar{\mathbf{f}}, \mathbf{X}, \bar{\mathbf{X}}) p(\bar{\mathbf{f}}|\bar{\mathbf{X}}) d\bar{\mathbf{f}} \propto \mathcal{N}(\boldsymbol{\Omega}^{-1} \boldsymbol{\kappa}|\mathbf{0}, \ \boldsymbol{\Lambda} + \mathbf{K}_{NM} \mathbf{K}_{MM}^{-1} \mathbf{K}_{MN}). \tag{23}$$

The full model marginal likelihood is given by:

$$\mathcal{L}_c^{ML}(\theta; D_c) = \sum_v \log \ p_\theta(\mathbf{y}_v|\mathbf{X}_v, \bar{\mathbf{X}}_v) = \sum_v \log \ \int p_\theta(\mathbf{y}_v|\boldsymbol{\omega}_v, \mathbf{X}_v, \bar{\mathbf{X}}_v) p(\boldsymbol{\omega}_v) d\boldsymbol{\omega}_v, \tag{24}$$

and the predictive distribution for a single data point $\mathbf{x}^*$ having the class $y^*$:

$$\mathcal{L}_c^{PD}(\theta; \mathbf{x}^*, y^*) = \sum_{v \in P^{y^*}} \log \ p_\theta(y_v^*|\mathbf{x}_v^*, \mathbf{y}_v, \mathbf{X}_v, \bar{\mathbf{X}}_v)$$

$$= \sum_{v \in P^{y^*}} \log \ \int p_\theta(y_v^*|\boldsymbol{\omega}_v, \mathbf{x}_v^*, \mathbf{y}_v, \mathbf{X}_v, \bar{\mathbf{X}}_v) p(\boldsymbol{\omega}_v|\mathbf{y}_v, \mathbf{X}_v) d\boldsymbol{\omega}_v.$$

$$= \sum_{v \in P^{y^*}} \log \ \int p(\boldsymbol{\omega}_v|\mathbf{y}_v, \mathbf{X}_v) \int p_\theta(y_v^*|f_v^*) p(f_v^*|\boldsymbol{\omega}_v, \mathbf{x}_v^*, \mathbf{y}_v, \mathbf{X}_v, \bar{\mathbf{X}}_v) df_v^* d\boldsymbol{\omega}_v. \tag{25}$$

To learn the model parameters, we first use block gibbs sampling with the posterior distributions $p(\mathbf{f}|\mathbf{y}, \boldsymbol{\omega}, \mathbf{X}, \bar{\mathbf{X}})$, and $p(\boldsymbol{\omega}|\mathbf{y}, \mathbf{f}) = PG(\mathbf{1}, \mathbf{f})$. Then, we use Fisher's identity [19] to obtain gradients w.r.t the model parameters with the marginal or predictive distributions.

Note that to speed up inference at test time, some computations that do not depend on $\mathbf{x}^*$ can be done offline. Importantly, we can sample and cache $\boldsymbol{\omega}$ and use it to compute $\boldsymbol{\Lambda}$ and the Cholesky decomposition of $\mathbf{B}$.

## C   Generalization bound - derivation

In Section 5 we presented an expression for the KL-divergence between the posterior and the prior (Eq. 10). Now we present the derivation:

$$KL[Q(\mathbf{f}) \| P(\mathbf{f})] = \int Q(\mathbf{f}) \log \frac{Q(\mathbf{f})}{P(\mathbf{f})} d\mathbf{f}$$

$$= \int Q(\mathbf{f}, \boldsymbol{\omega}) \log \frac{Q(\mathbf{f})}{P(\mathbf{f})} d\mathbf{f} d\boldsymbol{\omega}$$

$$= \int Q(\mathbf{f}, \boldsymbol{\omega}) \log \frac{Q(\mathbf{f}) Q(\boldsymbol{\omega}|\mathbf{f})}{P(\mathbf{f}) Q(\boldsymbol{\omega}|\mathbf{f})} d\mathbf{f} d\boldsymbol{\omega} \tag{26}$$

$$= \int Q(\mathbf{f}, \boldsymbol{\omega}) \log \frac{Q(\mathbf{f}|\boldsymbol{\omega})}{P(\mathbf{f})} d\mathbf{f} d\boldsymbol{\omega} + \int Q(\mathbf{f}, \boldsymbol{\omega}) \log \frac{Q(\boldsymbol{\omega})}{Q(\boldsymbol{\omega}|\mathbf{f})} d\mathbf{f} d\boldsymbol{\omega}$$

$$= \int Q(\mathbf{f}, \boldsymbol{\omega}) \log \frac{Q(\mathbf{f}|\boldsymbol{\omega})}{P(\mathbf{f})} d\mathbf{f} d\boldsymbol{\omega} + \int Q(\mathbf{f}, \boldsymbol{\omega}) \log \frac{Q(\mathbf{f}) Q(\boldsymbol{\omega})}{Q(\mathbf{f}, \boldsymbol{\omega})} d\mathbf{f} d\boldsymbol{\omega}$$

$$= \mathbb{E}_{Q(\boldsymbol{\omega})}\{KL[Q(\mathbf{f}|\boldsymbol{\omega})\|P(\mathbf{f})]\} - MI[\mathbf{f}; \boldsymbol{\omega}],$$

Where, the KL-divergence in the expectation term is between the prior, $P(\mathbf{f}) = \mathcal{N}(\boldsymbol{\mu}, \mathbf{K})$, and the posterior, $Q(\mathbf{f}|\boldsymbol{\omega}) = \mathcal{N}(\boldsymbol{\Sigma}(\mathbf{K}^{-1}\boldsymbol{\mu} + \boldsymbol{\kappa}), \boldsymbol{\Sigma})$. Therefore, it has the following closed-form:

$$KL[Q(\mathbf{f}|\boldsymbol{\omega})\|P(\mathbf{f})] = \frac{1}{2}\{log \frac{|\mathbf{K}|}{|\boldsymbol{\Sigma}|} - N_c + tr(\mathbf{K}^{-1}\boldsymbol{\Sigma}) + (\boldsymbol{\Sigma}(\mathbf{K}^{-1}\boldsymbol{\mu} + \boldsymbol{\kappa}) - \boldsymbol{\mu})^T \mathbf{K}^{-1} (\boldsymbol{\Sigma}(\mathbf{K}^{-1}\boldsymbol{\mu} + \boldsymbol{\kappa}) - \boldsymbol{\mu})\} \tag{27}$$

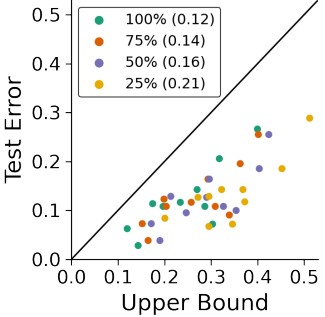

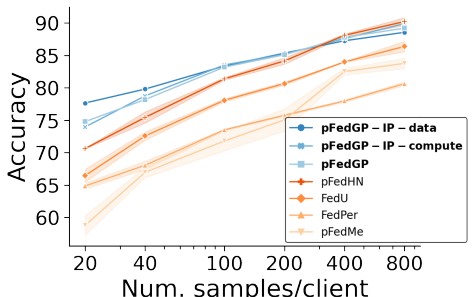

Figure 5: Test error vs. an estimated upper bound over 10 clients on CIFAR-10 with varying degrees of a training set data size using the Bayes classifier. Each dot represents a combination of client and data size. In parenthesis - the average difference between the empirical and the test error.

Figure 6: Model performance with varying degrees of an average number of training samples per client (x-axis in log scale). Results are over 50 clients on CIFAR-10.

# D   Experimental details

**Datasets.**   We evaluated pFedGP on the CIFAR-10, CIFAR-100 [38], and CINIC-10 [15] datasets. CIFAR-10/100 contain 60K images from 10/100 distinct classes, respectively, split to 50K for training and 10K for testing. CINIC-10 is a more diverse dataset. It is an extension of CIFAR-10 via the addition of down-sampled ImageNet [16] images. It contains 270K images split equally to train, validation, and test sets from 10 distinct classes.

**Data assignment.**   For partitioning the data samples across clients we followed the procedure suggested in [65, 72]. This procedure produces clients with a varying number of samples and a unique set of $k$ classes for each client. First, we sampled $k$ classes for each client. In general, we used $k = 2$ in CIFAR-10 experiments (e.g., Sections 5 & 6.1), $k = 10$ in CIFAR-100 experiments (e.g., Sections 6.1 & 6.2), and $k = 4$ in CINIC-10 experiments (e.g., Section 6.1). Next, to assign unique images for each client, we iterated over the classes and clients; for each client $i$ having the class $c$, we sampled an unnormalized class fraction $\alpha_{i,c} \sim U(.4, .6)$. We then assigned to the $i^{th}$ client $\frac{\alpha_{i,c}}{\sum_j \alpha_{j,c}}$ images from the overall samples of class $c$.

**Hyperparameter tuning.** We used a validation set for hyperparameter selection and early stopping in all methods. For the CIFAR datasets, we pre-allocated a validation set of size 10K from the training set. For the CINIC-10 dataset, we used the original split having a validation set of size 90K. The hyperparameters for all methods and all datasets were selected based on the learning setups with 50 clients. We searched over the learning-rates $\{1e-1, 5e-2, 1e-2, 5e-3\}$ for all methods, and personal learning-rates $\{5e-2, 1e-2, 5e-3, 1e-3\}$ for baseline methods only (pFedGP is a non-parametric approach and therefore does not optimize any private parameters). For pFedGP we searched over the number of epochs on sampled clients during training in $\{1, 3\}$. For baseline methods, since the training procedure varies significantly, we ran the baselines FOLA, LG-FedAvg, pFedMe, Per-FedAvg, FedU, and pFedHN according to the recommended configuration in their papers or code (which is usually a few epochs). Regarding FedAvg, FedPer, and pFedMe, we followed the protocol suggested by pFedME of using 20 iterations per client. For pFedGP We also searched over a scaling factor for the loss function in $\{1, 2\}$. We used the RBF kernel function with a fixed length scale of $1$ and an output scale of $8$. We used 20 parallel Gibbs chains for training and 30 parallel Gibbs chains for testing with 5 MCMC steps between samples in both. To compute the predictive distribution, at each tree node, we averaged over the log probabilities since it didn't impact the results but yielded a more calibrated model. In the reliability diagrams, we used 50 steps since as the number of steps increases usually the model becomes better calibrated (without a discernible accuracy change). In pFedGP-IP-data and pFedGP-IP-compute experiments, we used 100 inducing points per class. In all baselines that use FedAvg update rule, we used a variant of FedAvg in which a uniform weight was given to all clients during the global network update. All experiments were done on NVIDIA GeForce RTX 2080 Ti having 11GB of memory.

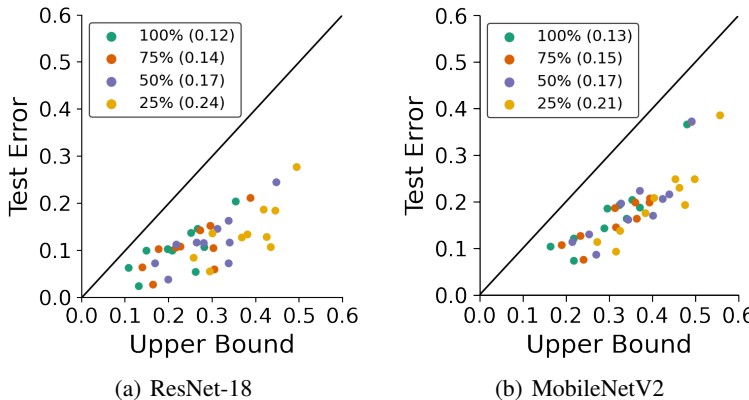

(a) ResNet-18                    (b) MobileNetV2

Figure 7: Test error vs. an estimated upper bound over 10 clients on CIFAR-10 with varying degrees of a training set data size on ResNet-18 (left) and MobileNetV2 (right). Each dot represents a combination of client and data size. In parenthesis - the average difference between the empirical and the test error.

**Noisy datasets (Sections 6.2 & E.3).** To generate a noisy version of the CIFAR-10 and CIFAR-100 datasets, we used the image-corruptions package [29]. We simulated the following 19 corruptions (Gaussian noise, shot noise, impulse noise, defocus blur, glass blur, motion blur, zoom blur, snow, frost, fog, brightness, contrast, elastic transform, pixelate, jpeg compression, speckle noise, Gaussian blur, spatter, saturate) with a corruption severity of (3, 4, 5), resulting in 57 unique noise models.

# E    Additional experiments

## E.1    Generalization bound - additional experiments

**Generalization bound with the Bayes risk.** In Section 5 we assessed the quality of the lower bound with the Gibbs risk. However, often we are interested in a deterministic predictor. Also, we would like to get an estimate of the error with a classifier that is closer to our estimate of $y^*$ with the Gauss-Hermite quadrature. This can be achieved with the Bayes risk [60] defined by $R_{Bayes}(Q) := \mathbb{E}_{(\mathbf{x}^*, y^*)}[sign\left(\mathbb{E}_{f^* \sim Q(f^*|\mathbf{x}^*, D_c)}[f^*]\right) \neq y^*]$. Figure 5 shows an estimation of the generalization error bound vs. the actual error on the novel clients with the Bayes classifier. From this figure we observe similar patterns to those seen in Figure 2. In general, the Bayes classifier performs better than the Gibbs classifier.

**Generalization with bigger networks.** To test how the bound behaves with larger networks, we also evaluated the Gibbs risk on ResNet18 [28] and MobileNetV2 [63], having $\sim$ 11.4M and $\sim$ 2.8M parameters correspondingly. In both networks, we replaced the final fully-connected layer with a linear layer of size 512 and we removed batch normalization layers. Results are presented in Figure 7. We observe a similar behavior with these networks to the one seen with the LeNet backbone. Namely, the bound is data-dependent and gives non-vacuous guarantees.

## E.2    Varying the training set size

To decouple the effect of the local dataset size from the number of clients in the system, we altered the setting in Section 6.1. Here, we fixed the number of clients to $50$ and used stratified sampling to sample $\{1000, 2000, 5000, 10000, 20000, 40000\}$ examples from the training dataset of CIFAR-10, where $40000$ is the total number of training examples. Then, similar to Section 6.1 we randomly assigned two classes per client and partitioned the (sampled) data across clients. Figure 6 shows that all methods suffer from accuracy degradation when the training data size decrease; however, in pFedGP methods, the reduction is less severe compared to baseline methods. We especially note pFedGP-IP-data which shows remarkable accuracy in the extremely low data regime (77.7% accuracy with only 1000 training examples). These results can be attributed to the sharing of the inducing

Table 3: Test accuracy (± SEM) over 100 clients on noisy CIFAR-10 under a homogeneous class distribution.

| Method | Accuracy |
|---|---|
| FedAvg [51] | 42.0 ± 0.2 |
| FedPer [4] | 38.2 ± 2.0 |
| LG-FedAvg [44] | 41.5 ± 0.2 |
| pFedme [72] | 36.3 ± 0.0 |
| FedU [18] | 24.8 ± 0.0 |
| pFedHN [65] | 35.7 ± 0.3 |
| **Ours** | |
| pFedGP-IP-data | 46.1 ± .05 |
| pFedGP-IP-compute | 46.5 ± 0.5 |
| pFedGP | **46.7 ± 0.2** |

Table 4: Test accuracy (± SEM) over 50, 100 clients on CIFAR-10 under a homogeneous class distribution.

| Method | # clients | |
|---|---|---|
| | 50 | 100 |
| FedAvg [51] | **66.2 ± 0.3** | 64.6 ± 0.2 |
| FedPer [4] | 56.8 ± 0.1 | 50.9 ± 0.4 |
| pFedMe [72] | 46.9 ± 0.5 | 44.4 ± 0.3 |
| FedU [18] | 29.6 ± 0.6 | 26.3 ± 0.5 |
| pFedHN [65] | 62.8 ± 0.5 | 56.5 ± 0.1 |
| **Ours** | | |
| pFedGP-IP-data | 62.7 ± 0.7 | 62.4 ± 0.5 |
| pFedGP-IP-compute | 65.8 ± 0.3 | 65.2 ± 0.4 |
| pFedGP | **66.2 ± 0.3** | **65.7 ± 0.2** |

Table 5: pFedGP-IP-compute vs. pFedGP full model test accuracy and average predictive posterior inference run-time (± STD) as a function of the number of inducing points (IPs) over 50 clients on CINIC-10.

| Num. IPs | 80 | 160 | 240 | 320 | 400 | ǀ | Full GP |
|---|---|---|---|---|---|---|---|
| Accuracy (%) | 70.2 | 71.0 | 71.4 | 71.5 | 71.5 | ǀ | 71.3 |
| Run time (sec.) | 0.15 ± .02 | 0.21 ± .03 | 0.27 ± .04 | 0.34 ± .05 | 0.42 ± .06 | ǀ | 1.08 ± .16 |

inputs which effectively increase the size of the training data per client. The LG-FedAvg baseline was excluded from this figure since it showed low accuracy.

### E.3 PFL with input noise under a homogeneous class distribution

In Section 6.2 we evaluated pFedGP under two types of variations between the clients: (i) a unique input noise, and (ii) in the class distribution. It was done on the CIFAR-100 dataset with 100 clients. Here, we consider only a shift in the input noise between clients while having a balanced class distribution in all clients. We do so on the CIFAR-10 dataset (i.e., each client has 10 classes distributed approximately equal). Similarly to Section 6.2 we configured 100 clients, distributed the data among them, and assigned a noise model to each client from a closed-set of 57 noise distributions. Table 3 shows that in this setting as well, pFedGP and its variants achieve high accuracy and surpass all baseline methods by a large margin. From the table, we also notice that FedAvg, which performs well in balanced class distribution setups, outperforms all competing methods except ours.

### E.4 Homogeneous federated learning with CIFAR-10

pFedGP is a non-parametric classifier offered for personalized federated learning. Therefore, it only needs to model classes that are present in the training set. Here, for completeness, we evaluate pFedGP against top-performing baselines on the CIFAR-10 dataset, where all classes are represented in all clients. We do so on a homogeneous federated learning setup, i.e., all classes are distributed equally across all clients. Data assignment was done similarly to the procedure described in Section 6.1 (see Section D for more details). Table 4 shows that pFedGP outperforms all PFL baseline methods by a large margin in this setting as well. An interesting, yet expected, observation from the table is that FedAvg performs well under this (IID) setup. This result connects to a recent study that suggested that under a smooth, strongly convex loss when the data heterogeneity is below some threshold, FedAvg is minimax optimal [11]. We note here that modeling classes that are not present in the training set with pFedGP can be accomplished easily with one of the inducing points variants of pFedGP.

Table 6: pFedGP model variants test accuracy ($\pm$ SEM) over 50, 100, 500 clients on CIFAR-10, CIFAR-100, and CINIC-10. The *# samples/client* indicates the average number of training samples per client.

| | CIFAR-10 | | | CIFAR-100 | | | CINIC-10 | | |
|---|---|---|---|---|---|---|---|---|---|
| # clients | 50 | 100 | 500 | 50 | 100 | 500 | 50 | 100 | 500 |
| # samples/client | 800 | 400 | 80 | 800 | 400 | 80 | 1800 | 900 | 180 |
| pFedGP-IP-data w/o personalization | 88.6 ± 1.0 | 87.0 ± 0.2 | 86.4 ± 0.7 | 58.1 ± 0.3 | 57.4 ± 0.6 | 55.4 ± 0.2 | 69.2 ± 0.3 | 68.2 ± 0.9 | 67.9 ± 0.1 |
| pFedGP-IP-data | 88.6 ± 0.2 | 87.4 ± 0.2 | 86.9 ± 0.7 | 60.2 ± 0.3 | 58.5 ± 0.3 | **55.7 ± 0.4** | 69.8 ± 0.2 | 68.3 ± 0.6 | 67.6 ± 0.3 |
| pFedGP-IP-compute-marginal | **89.8 ± 0.6** | **88.8 ± 0.3** | **87.8 ± 0.3** | 60.9 ± 0.4 | 58.8 ± 0.3 | 46.7 ± 0.3 | **72.1 ± 0.2** | 71.1 ± 0.6 | 67.5 ± 0.2 |
| pFedGP-IP-compute-predictive | **89.9 ± 0.6** | **88.8 ± 0.1** | 86.8 ± 0.4 | 61.2 ± 0.4 | 59.8 ± 0.3 | 49.2 ± 0.3 | **72.0 ± 0.3** | **71.5 ± 0.5** | 68.2 ± 0.2 |
| pFedGP-marginal | 89.0 ± 0.1 | 88.0 ± 0.2 | 86.8 ± 0.2 | **63.7 ± 0.1** | **61.4 ± 0.3** | 50.3 ± .05 | 71.6 ± 0.3 | 71.0 ± 0.6 | **68.5 ± 0.2** |
| pFedGP-predictive | 89.2 ± 0.3 | **88.8 ± 0.2** | 87.6 ± 0.4 | 63.3 ± 0.1 | **61.3 ± 0.2** | 50.6 ± 0.2 | 71.8 ± 0.3 | 71.3 ± 0.4 | 68.1 ± 0.3 |

## E.5 Computing demands

In this section we evaluate pFedGP computational requirements. First we compared between pFedGP full model (Section 4.1) and pFedGP-IP-compute (Section 4.3) in terms of accuracy and run-time during test time. The key component controlling the computational demand of pFedGP during test time is the predictive distribution (Equations 17 & 22). After the training phase, when a new test sample arrives, computing the predictive distributions can be done efficiently by using cached components that depend only on the training data (e.g., the Cholesky decomposition of $\mathbf{B}$) and can be calculated offline. Therefore, to quantify the impact of using pFedGP-IP-compute compared to the full GP model, we recorded in Table 5 the federated accuracy and average time per client for calculating the predictive distribution for all test examples as a function of the number of inducing points. The comparison was done on the pre-allocated test set from the CINIC-10 dataset over 50 clients (i.e., $\sim 1800$ test examples per client divided to 4 classes) using a model that was trained with 100 inducing points. The table shows a significant improvement in the run time compared to the full model without any (or only minor) accuracy degradation. We note here that including the network processing time will add a constant factor of 0.03 seconds.

In addition to the above test, we also tracked pFedGP full model and baseline methods memory usage and runtime on CIFAR-10 and CIFAR-100 with 50 clients during training. For comparability, we fixed the number of epochs that each sampled client makes to one for all methods. We found that pFedGP computational requirements are reasonable for running it in current FL systems [9]. pFedGP needed $\sim 1.4/1.6$ GB memory, and a run-time of $\sim 1/2$ hours for CIFAR-10/100 correspondingly. Baseline methods needed 1.1-1.3 GB memory and took $\sim 25$ minutes to run on both datasets. According to this naive testing, pFedGP is computationally more intensive compared to standard methods such as FedAvg. However, in return for that additional complexity, it obtains substantial performance gains compared to the baseline methods. Furthermore, when implementing pFedGP there are some trade-offs, for example, iterating over the tree can be either sequential to obtain lower memory cost or parallelized to obtain shorter running times. Or, when applying the Gibbs sampling, the complexity can be affected by the number of parallel Gibbs chains and the number of MCMC steps which constitute a trade-off between accuracy and runtime. Finally, note that using only one epoch of training damaged severely the performance of several baseline methods. Therefore, to obtain the same accuracy as reported in this paper the gaps in run-time are actually smaller.

## E.6 Meta-Learning approaches for FL

In this study we advocate the use of GPs in general, and pFedGP specifically, in FL systems. A key motivation for using GPs is that often the data on clients is limited. Alternatively, we could have used other methods that were found to work well with limited data. Specifically, methods that are based on the model-agnostic meta-learning (MAML) [43] learning procedure. Here, we compare pFedGP against the MAML-based federated learning approach *Per-FedAvg* [21] under the setting presented in Section 6.1 in the main text. We observed the following results: $83.5 \pm .05/81.7 \pm 0.4/76.4 \pm 1.0$ on CIFAR-10, and $45.6 \pm 0.2/41.0 \pm 1.4/30.2 \pm .02$ on CIFAR-100 with 50/100/500 clients respectively. Comparing this method to pFedGP reveals that our approach outperforms this baseline as well.

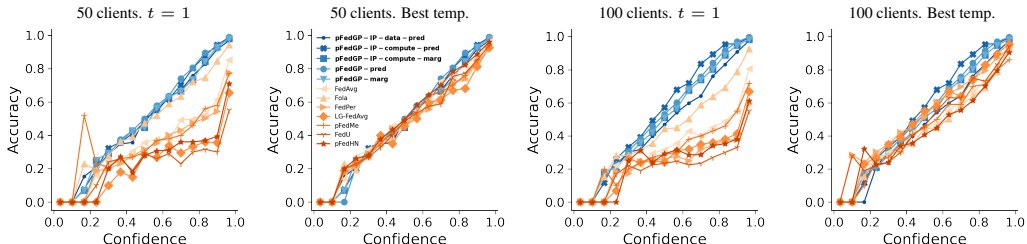

Figure 8: Reliability diagrams over 50, 100 clients on CIFAR-100.

### E.7 Predictive vs marginal likelihood

In the main text, we presented two alternatives for learning the model parameters with pFedGP and pFedGP-IP-compute, the predictive distribution, and the marginal likelihood (see Section 4). We now compare between these two alternatives in Table 6 under the standard setup presented in Section 6.1. The table shows that for both pFedGP and pFedGP-IP-compute the two variants are comparable with a slight advantage to the predictive distribution objective. Nevertheless, using the marginal likelihood usually results in a better-calibrated model (Section E.9).

### E.8 pFedGP-IP-data Ablation

Recall that for the pFedGP-IP-data variant during training we build the kernel with the (shared) inducing inputs only. Yet, during test time, to account for the personal data, we use both the inducing inputs and the training data of the client for building the kernel. This method is especially effective in cases where the data per client is limited. Here, we evaluate this method without using the actual training data during test time. This means that the only personalization derives from the personal tree structure that is formed based on the actual training data of the client. Remarkably, Table 6 shows that using this strategy yields high accuracy as well and it is often comparable to pFedGP-IP-data.

In pFedGP-IP-data we also introduced a correction term based on the class probabilities. Here we investigated the impact of this functionality as well. When the data is distributed uniformly among the client's classes, this correction term does not have any effect. Therefore, we tested its effect under a similar setting to the one presented in Section 6.3 on CIFAR-10. Namely, we sampled examples from classes according to a Dirichlet distribution with parameter $\alpha = 0.1$ for each client. With the class balancing, we noticed an accuracy of $84.4 \pm 0.5$ without it the accuracy dropped to $83.7 \pm 0.5$.

### E.9 Reliability diagrams

Now we present additional reliability diagrams for CIFAR-100 with 50 and 100 clients, with and without temperature scaling (See Figure 8 for unified diagrams and Figures 9 - 12 for separate diagrams). For each baseline we applied a grid search over a temperature $t \in \{0.1, 0.2, 0.5, 1.0, 2.0, 5.0, 10.0, 20.0, 50.0, 100.0, 200.0, 500.0, 1000.0\}$, chose the best temperature based on the pre-allocated validation set according to the ECE, and generated the diagram using the test set data. In addition, we present here reliability diagrams obtained by optimizing the marginal likelihood for pFedGP and pFedGP-IP-compute. From the figures, pFedGP does not gain from temperature scaling as baseline methods do since it is a calibrated classifier by design. Although this procedure improves the calibration of baseline methods, we note that finding the right temperature requires having a separate validation set which often can be challenging to obtain for problems in the low data regime.

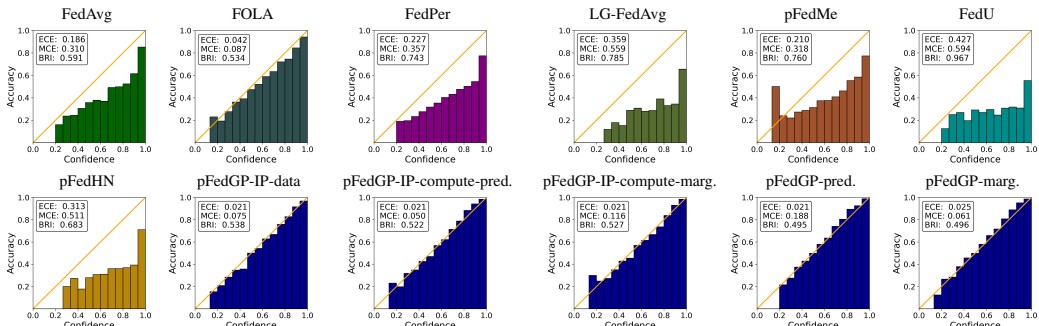

Figure 9: Reliability diagrams on CIFAR-100 with 50 clients. Default temperature ($t = 1$). The last 5 figures are ours.

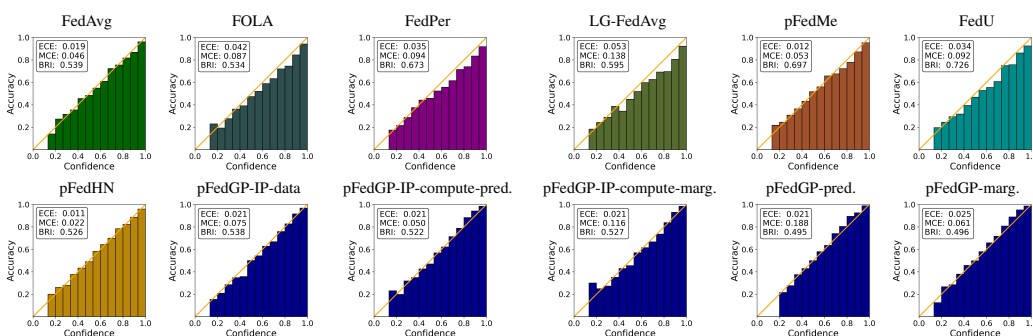

Figure 10: Reliability diagrams on CIFAR-100 with 50 clients. Best temperature. The last 5 figures are ours.

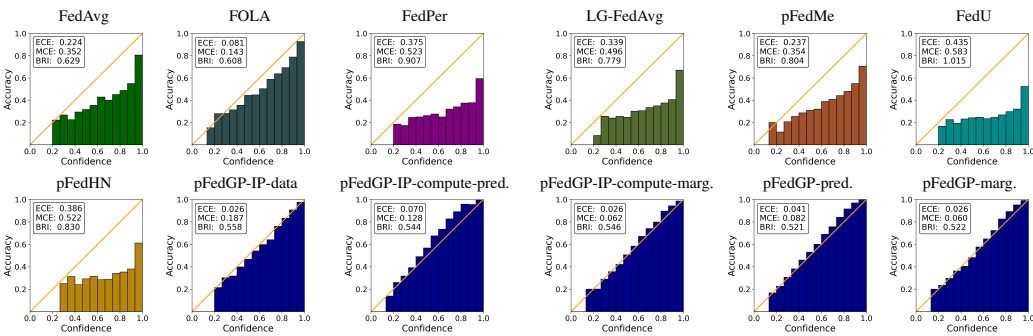

Figure 11: Reliability diagrams on CIFAR-100 with 100 clients. Default temperature ($t = 1$). The last 5 figures are ours.

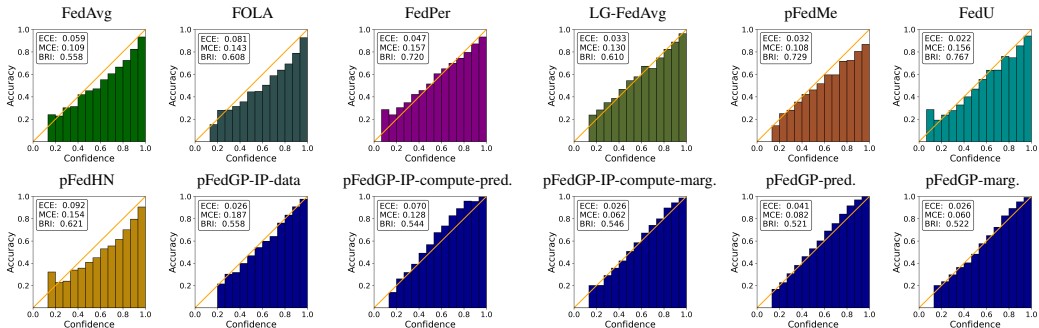

Figure 12: Reliability diagrams on CIFAR-100 with 100 clients. Best temperature. The last 5 figures are ours.