# OpenReview forum: "Personalized Federated Learning With Gaussian Processes"
_NeurIPS.cc/2021/Conference — NeurIPS 2021 Poster_

### Official Review · Reviewer_anUe · 2021-07-05

**Rating:** 7
**Confidence:** 4

**Summary:**

This paper presents a new framework for personalized federated learning via Gaussian process classification (GPC) model. The key idea is to have each client node maintain its own GPC model but share the NN feature map f(x). Here, the GPC model of client c is equipped with kernel k_c(x, x') = k_c( f(x), f(x'); w_c)  with personalized parameters w_c.

Thus, per iteration, a subset of clients is selected. Each client c downloads the shared f(x) from server and re-build its GPC model (i.e., re-fitting for w_c) using local data D_c. A local updated feature map f(x) (via gradient propagation) is then computed for each client and communicated to the server, which averages the updated model to form a new feature map f(x).

The proposed framework is evaluated on 3 image datasets: CIFAR-10, CIFAR-100 and CINIC-10. The reported performance shows that it outperforms a selected set of baselines substantially when the amount of data per client is small & heterogeneous.

**Ethical Concerns:**

No concern.

**Limitations And Societal Impact:**

The authors have included a broader impact statement elaborating on the societal impact of the proposed work. I find that acceptable & have no further concerns.

**Main Review:**

NOVELTY & SIGNIFICANCE

I believe this is the first personalized federated learning framework that emphasizes on separating between learning the model (which is private) and learning feature representation (which is shared and can be learned via FL). This is an inherent separation the construction of a GP model which makes it a good model base. In addition, choosing GP as a base model is also a practical choice seeing that GP is often more effective on low-data regime (thus, being more suitable to provide personalized performance to client with little data). Thus, concept-wise, I consider the proposed method novel.

On the other hand, the proposed method appears to combine straight-forwardly a bunch of existing techniques (i.e., a vanilla local GP training method, direct adoption of FedAvg to learn a shared feature map & a vanilla PAC bound to assess the generalized performance on novel clients) so algorithmic-wise, there is little novelty & significance. That being said, I also believe the significance of the proposed method is more on the practical side, regarding its extensive empirical studies, on which I will further comment below.

QUALITY

The paper is technically correct. I have gone through its technical development & derivation and have not found any flaws. I do, however, find the contributions of the paper somewhat orthogonal at certain parts:

First, it is not clear to me the role of a GP-tree model in this context -- I thought any GPC would do so if we are to pick a more sophisticated model, there must be a reason. I guess it has to do with the performance but I am not sure why so could the author elaborate more on this?

Likewise, I also do not see the advantage of using Polya-Gamma augmentation over normal approximation techniques for GPC. Please also discuss this in the rebuttal.

---

Regarding the experiments, I am a bit confused that pFedGP-IP-data is inferior to pFedGP-IP-compute. This is somewhat strange since pFedGP-IP-compute only uses the shared inducing points so there is not much to personalize. In contrast, pFedGP-IP-data includes both inducing points and local data points for inference so it is not clear to me why pFedGP-IP-data is inferior to pFedGP-IP-compute. Could the authors elaborate more on this too?

In addition, I find the personalized FL work in [16] very relevant & is directly applicable to the setting here. Could the author provide extra, minimal comparison with [16] in the rebuttal?

---

Regarding the theory, I suppose the point of section 5 is that if we use the feature map directly to build a personalized GPC to a novel client, we can make use of the PAC-bound for GPC in [56] to (empirically) measure the generalization gap. While this is not exactly a novel point, I appreciate the empirical analysis that makes use of this bound to showcase the enhanced OOD generalizability of the proposed model on novel clients, especially there is an increasing distributional shift. This speaks well to the practical significance of the proposed method.

CLARITY

The paper is very well-written. However, I feel that to a more general audience, the lack of background cover on Polya-gamma & GP-tree might impair readability. I'd suggest cutting down on section 5 to make room for those background detail.

REVIEW SUMMARY

In general, I enjoy reading this paper. The proposed method provides quite a refreshing perspective and it has also been evaluated meticulously. That being said, I do have some questions for the authors & some of which is rather important (e.g., comparison with [16]) and I hope the authors would address those in the rebuttal.

**Time Spent Reviewing:**

6

---

> ### Author Response · Authors · 2021-08-10
> **Response to Reviewer anUe**
>
> We thank the reviewer for the thoughtful comments. Please see our responses below.
>
> **1**
> > “[...] the proposed method appears to combine straight-forwardly a bunch of existing techniques [...] so algorithmic-wise, there is little novelty & significance. [...]”
>
> We kindly disagree with the claim that there is a little novelty & significance. Indeed in the paper we built upon existing GP techniques. However, we introduced both conceptual ideas, such as the shared learning of the kernel and the sharing of the inducing inputs, and non-trivial modifications to existing techniques.
>
> To obtain the results in the paper we introduced several novelties that, to the best of our knowledge, weren’t presented before:
> 1. We presented an alternative learning scheme of the model parameters with GP-Tree based on a Gibbs sampling approach that is more flexible and is able to quantify uncertainty more reliably.
> 2. To the best of our knowledge, we are the first to propose the inducing point method in section 4.2 and the class distribution correction for it.
> 3. We are the first to show how FITC can be combined with a Polya-Gamma based classifier to obtain a more scalable classifier and how to learn with it the model parameters using MCMC samples for classification tasks.
> 4. We were also the first to adapt the PAC Bayes bound to Polya-Gamma classifiers and provide an estimation of the bound. Although this result is general it is very relevant to the scenario of having a new client entering the system.
>
> We believe that these novelties accumulate to a novel, general, and complete solution for the FL setup. As an indication, as the reviewer stated, our method outperforms by a large margin other baselines in a variety of settings and on several datasets.
>
> **2**
> > “[...] it is not clear to me the role of a GP-tree model in this context -- I thought any GPC would do so if we are to pick a more sophisticated model, there must be a reason. [...] Likewise, I also do not see the advantage of using Polya-Gamma augmentation over normal approximation techniques for GPC.”
>
> Our method is somewhat general, and indeed other GPC could have been chosen. However, we picked GP-Tree because of the following three main reasons:
> 1. As the reviewer stated, because of its good predictive abilities. In [1] it was shown that GP-Tree outperforms common GP classification methods, ones that use the Polya-Gamma augmentation and ones that don’t. Therefore, it was a natural choice for this setup.
> 2. Standard GPC methods use variational approximation. As a result, they may underestimate the variance and generate overconfident predictions [2, 3]. On the other hand, with the Polya-Gamma augmentation we can use Gibbs sampling that results in a more accurate uncertainty quantification and has asymptotic guarantees.
> 3. Using GP-Tree allows controlling the computational demand. The memory and run-time can be controlled by adjusting the number of parallel Gibbs chains and the number of MCMC steps. In our experiments we noticed that using several chains (>5) usually results in a higher accuracy (~2% difference compared to only one chain), while applying more Gibbs steps usually doesn’t affect the accuracy but does increase the uncertainty quantification (up to 0.02 units in the ECE compared to doing only one step).
>
> **3**
> > “Regarding the experiments, I am a bit confused that pFedGP-IP-data is inferior to pFedGP-IP-compute. This is somewhat strange since pFedGP-IP-compute only uses the shared inducing points so there is not much to personalize. In contrast, pFedGP-IP-data includes both inducing points and local data points for inference so it is not clear to me why pFedGP-IP-data is inferior to pFedGP-IP-compute.”
>
> This is a great question. We will make this point clearer in the revised version of the paper. In pFedGP-IP-compute the training data takes an active part in the GP inference formulas (Eq. 6 & 7). Therefore, the data is a key factor in the model construction and as a result in the personalization. On the other hand, in pFedGP-IP-data when training the model the actual data impact in a weak manner only through the loss function (L187). As a result, we believe that the discriminative power of the learned kernel (i.e., the underlying neural network) for pFedGP-IP-data is not as strong as pFedGP-IP-compute.
>
> **4**
> > “I find the personalized FL work in [16] very relevant & is directly applicable to the setting here. Could the author provide extra, minimal comparison with [16] in the rebuttal?”
>
> Thank you for focusing our attention to (Fallah at el., 2020). We add here comparison to that baseline in the table below. The table shows the test accuracy of this method on CIFAR-10 and CIFAR-100 with 50 and 100 clients. Comparing this method to pFedGP reveals that our approach outperforms this baseline as well.
>
> . | CIFAR-10 | CIFAR-10 | CIFAR-100 | CIFAR-100
> ----------|--------|--------|--------|--------
> Num. clients |	50 |    100 |  50  |    100
> Per-FedAvg | 83.6±0.5 | 82.3±0.2 | 45.4±0.8  | 47.9±0.7
> pFedGP | **89.2±0.3** | **88.8±0.2** | **63.3±0.1** | **61.3±0.2**
>
> **5**
> > The paper is very well-written. However, I feel that to a more general audience, the lack of background cover on Polya-gamma & GP-tree might impair readability”
>
> Thank you. We added an extensive background on the Polya-Gamma augmentation and GP-Tree in the Appendix. We will reorganize the paper so it will be more readable to the general audience.
>
> [1] Achituve, I., Navon, A., Yemini, Y., Chechik, G., & Fetaya, E. (2021). GP-Tree: A Gaussian Process Classifier for Few-Shot Incremental Learning. arXiv preprint arXiv:2102.07868.
>
> [2] Knoblauch, J., Jewson, J., & Damoulas, T. (2019). Generalized variational inference: Three arguments for deriving new posteriors. arXiv preprint arXiv:1904.02063.
>
> [3] Minka, T. (2005). Divergence measures and message passing. Technical report, Microsoft Research.

---

> > ### Comment · Reviewer_anUe · 2021-08-20
> > **Response to Author's rebuttal**
> >
> > Thank you for the detailed response & the additional results. The rebuttal has addressed my concerns sufficiently.
> >
> > I am still not entirely clear why Polya-Gamma augmentation would induce a better uncertainty quantification but that is not exactly the key point here so that is of a more minor concern. But, please do consider elaborating more on this in the revised version.
> >
> > Please also include the extra results with (Fallah et al., 2020).

---

> > > ### Author Response · Authors · 2021-08-21
> > > **Response to Reviewer anUe Comment**
> > >
> > > Thank you. We will include the additional results as well as other comments that were raised in the revised version of the paper. We will also further elaborate on the predictive advantage of the Polya-Gamma augmentation scheme. In general, the Polya-Gamma augmentation benefits from fast mixing and has the ability of even a single value of $\omega$ to capture much of the volume of the marginal distribution over function values. This phenomenon can be observed in [4], Appendix B (ArXiv version) or Appendix C (NeurIPS version).
> > >
> > > [4] Linderman, S., Johnson, M. J., & Adams, R. P. (2015). Dependent Multinomial Models Made Easy: Stick-Breaking with the Polya-gamma Augmentation. Advances in Neural Information Processing Systems, 28, 3456-3464.

---

### Official Review · Reviewer_NyvV · 2021-07-12

**Rating:** 6
**Confidence:** 4

**Summary:**

This work considers the task of learning user personalised predictive models in the federated learning setting. It is realised via a local Gaussian Process (GP) on each client which operates on top of features extracted from a neural network that is shared across the entire federation, named pFedGP. The authors motivate the usage of GPs via their Bayesian nature which allows for good performance in the low data regime (common in federated learning) as well as the extraction of prediction probabilities that are calibrated (crucial for safety-critical applications of FL).

Since the authors operate on federated classification tasks,  they employ the GP classification model proposed at [1], which decomposes a multi-class prediction problem into a series of binary predictions. The benefit of this is that one can use the Polya-Gamma augmentation scheme, which can allow for efficient sampling of the posterior distributions, necessary for learning the hyper parameters of the model. The authors then  introduce two additional variants of their overall model which employ inducing points, a common technique in the GP literature. The first one is motivated for improving the performance of the model in the case of limited data, whereas the second one is about making the model more scalable in the case of large local datasets. Finally, the authors also provide a PAC-Bayesian generalization bound for the model performance one new clients that enter the federation.

The paper concludes with a series of experiments that compare the performance of pFedGP against several baselines for federated learning and personalised federated learning.

[1] GP-Tree: A Gaussian Process Classifier for Few-Shot Incremental Learning, Achituve et al., 2021

**Limitations And Societal Impact:**

Yes.

**Main Review:**

This work tackles the problem of personalisation in federated learning which is an important and active area of research in federated learning. The parts of the proposed approach are not novel themselves, however their combination for the task of personalised FL is new. The discussion of the method is relatively clear, and it is nice to see that the uncertainties provided by pFedGP are well calibrated.

Despite these positive aspects, I cannot recommend for accepting the paper as is currently and below I elaborate why:
- The overall method seems quite computationally intensive, as one needs to perform clustering and several iterations of Gibbs sampling in order to learn the model. Since the compute is done locally in resource constrained devices in federated learning, I believe it is important for the authors to contrast the computational requirements (e.g., in terms of floating point operations) of their method against the more traditional NN training of, e.g., FedAvg. Furthermore, it would also be helpful if the authors include a pseudo-code of the overall federated training procedure with the different variants of pFedGP.
- The generalisation bound provided in the submission is independent of a core part of the model, the neural net. As a result, one can wonder about its usefulness in practice where a wide variety of neural networks might be employed. For the current experiments it seems to work reasonably well, however the network used is a simple LeNet and the datasets (CIFAR10/100, CINIC-10) are quite  for the complex for this architecture (so it is harder to overfit). In an ideal case, it would be good if the authors include the neural network in the generalisation bound. As this might be harder to do, it would perhaps be worthwhile if the authors can show the behaviour of the bound with more complicated networks that have more parameters. Furthermore, there are some approximations involved in computing the bound with the Polya-Gamma augmentation (i.e., the expectation over $Q(w)$ and estimation of the mutual information). It is a bit unclear how finite-sample estimates of these quantities affect the actual bound and, ideally, they should be taken into account when providing the actual guarantee.
- The experimental section and setup is a bit unclear with respect to a couple of important FL training hyperparameters. The authors use non-iid settings for the experiments. In prior work [2], it has been shown that in such scenarios, doing less epochs of local optimization (denoted as $E$) and using more clients per round can be beneficial for the final performance of the model when using FedAvg (which makes sense, as more local epochs can push each of the local models further away from each other, as each client has its own specialised dataset). Going over the authors’ appendix, it seems that for the baselines they use $E \approx 20$ but $E \in set(1, 3)$ for pFedGP. In fact, the reference [2] has managed to train a Lenet type of network with FedAvg on a similar non-iid version of CIFAR 10  and got up to ~73% performance, which is much better than the ~60% this work reports. It would be good if the authors adapt the hyperparameters appropriately so that the comparison is more meaningful. Finally, for a better personalised model performance baseline that is quite simple you should consider the performance of the locally fine-tuned model when training with FedAvg, see [3].
- One result that was relatively surprising was that FedPer had much worse performance than pFedGP, as they are quite similar in a sense; they both use the same backbone, except that one uses a linear classifier whereas the other uses a GP. It would be great if the authors can explain why this is the case. Furthermore, upon inspecting the author’s code it seems that they use two convolutional followed by two fully connected layers + one extra embedding layer, before feeding the embeddings to the GP. This is a larger model than the traditional lenet which, after the two convolutional layers, uses a single fully connected  layer and then the linear classifier. What kind of backbone did the authors use for their baselines? Did it have the same number of layers? (Unfortunately, I couldn’t find the code for the baselines in the supplementary). I would suggest that, for future comparisons, the authors, use a comparable architecture to lenet, in that they use the output of the first fully connected layer as an embedding for the GP.

Finally, below I have some further comments about the clarity of this work and how it could be improved:
- At line 192 in the main text the authors mention that inducing points can distort the true class distribution for pFedGP and that they use reweighting to combat this effect. In order to improve the clarity of the submission, I believe that the authors should provide an example of this event and why it happens. Furthermore, it would be beneficial if they explain how critical is this modification and how it is done in practice (i.e., what is $q(y, x)$, $p(y, x)$?).
- On lines 301-302 the authors mention that pFedGP-data is especially helpful when few datapoints per class are available, nevertheless, this seems to be the case only for CIFAR100 but not for CIFAR10 / CINIC-10. It would be helpful in the authors explain why. Finally, it seems that the inducing points were fixed to 100 for all the experiments? How critical of a hyper parameter is the number of inducing points for the performance of pFedGP?


[2] Measuring the Effects of Non-Identical Data Distribution for Federated Visual Classification, Hsu et al., 2019

[3] Improving Federated Learning Personalisation via Model Agnostic Meta Learning, Jiang et al., 2019

**Time Spent Reviewing:**

7 hours

---

> ### Author Response · Authors · 2021-08-10
> **Response to Reviewer NyvV - Part 1**
>
> We thank the reviewer for the thoughtful comments. Please see our responses below.
>
> **1**
> > “[...] The parts of the proposed approach are not novel themselves [...]”
>
> We kindly disagree with the claim that the parts of the proposed approach are not novel. Indeed in the paper we built upon existing GP techniques. However, we introduced both conceptual ideas, such as the shared learning of the kernel and the sharing of the inducing inputs, and non-trivial modifications to existing techniques.
>
> To obtain the results in the paper we introduced several novelties that, to the best of our knowledge, weren’t presented before:
> 1. We presented an alternative learning scheme of the model parameters with GP-Tree based on a Gibbs sampling approach that is more flexible and is able to quantify uncertainty more reliably.
> 2. To the best of our knowledge, we are the first to propose the inducing point method in section 4.2 and the class distribution correction for it.
> 3. We are the first to show how FITC can be combined with a Polya-Gamma based classifier to obtain a more scalable classifier and how to learn with it the model parameters using MCMC samples for classification tasks.
> 4. We were also the first to adapt the PAC Bayes bound to Polya-Gamma classifiers and provide an estimation of the bound. Although this result is general it is very relevant to the scenario of having a new client entering the system.
>
> We believe that these novelties accumulate to a novel, general, and complete solution for the FL setup. As an indication, our method outperforms by a large margin other baselines in a variety of settings and on several datasets.
>
> **2**
> > “The overall method seems quite computationally intensive, as one needs to perform clustering and several iterations of Gibbs sampling in order to learn the model. [...], I believe it is important for the authors to contrast the computational requirements of their method against the more traditional NN training of, e.g., FedAvg.”
>
> Reviewer *ys9C* had a similar concern. We will share the same answers for both. pFedGP is indeed more computational intensive compared to standard methods such as FedAvg. In return to that additional complexity we were able to obtain substantial performance gains compared to the baseline methods. When implementing pFedGP there are some trade-offs, for example, iterating over the tree can be either sequential to obtain lower memory cost or parallelized to obtain shorter running times. When applying the Gibbs sampling, the complexity can be affected by the number of parallel Gibbs chains and the number of MCMC steps which constitute a tradeoff between accuracy and runtime. In fact, in our experiments we noticed a reasonable drop in accuracy (~2% at most) when using only one chain and taking a few MCMC steps.
>
> Regarding the clustering algorithm, since the k-means algorithm is applied on the class prototypes only (one representative example for each class), the number of data points is small and in the worst case is at the size of the number of classes.
>
> Following this comment, we tracked pFedGP and baseline methods memory usage and runtime on CIFAR-10 and CIFAR-100 with 50 clients. For comparability, we fixed the number of epochs that each sampled client makes to one. pFedGP needed ~1.4/1.6 GB memory, and a run-time of ~1/2 hours for CIFAR-10/100 correspondly. Baseline methods needed 1.1-1.3 GB memory and took ~25 minutes to run on both datasets. We believe pFedGP computational requirements are reasonable for running it in current FL systems [1] and considering that the code is highly not optimized.
>
> Also, when considering only the inference time we can obtain additional speed up by precomputing and caching many of the computations (e.g. Cholesky decomposition). Baseline method needed ~0.6ms to run a single example while pFedGP needed ~6ms for CIFAR-10 and ~30ms for CIFAR-100. While relatively much slower, it is still quite fast and can run in real-time.
>
> Finally, we note that while scaling GPs is an open research problem [2, 3] and many advances can be used to scale our method, our main focus is the low data regime where these issues are not as restrictive.
>
> **3**
> > “it would also be helpful if the authors include a pseudo-code of the overall federated training procedure with the different variants of pFedGP.”
>
> Thank you for this suggestion. We will add a pseudo-code of pFedGP variants to the revised version of the paper.
>
> **4**
> > “The generalisation bound provided in the submission is independent of a core part of the model, the neural net. [...]. For the current experiments it seems to work reasonably well, however the network used is a simple LeNet [...]. In an ideal case, it would be good if the authors include the neural network in the generalisation bound. As this might be harder to do, it would perhaps be worthwhile if the authors can show the behaviour of the bound with more complicated networks that have more parameters.”
>
> Thank you for pointing it out. Following your suggestion we evaluated the bound with two additional backbone networks ResNet18 and MobileNetV2, having ~11.4M and ~2.8M parameters correspondingly under the same experimental setup of Section 5. In both networks we replaced the final fully connected layer with a linear layer of size 512. Also, since incorporating batch normalization layers in federated learning systems is an open research problem [4 ,5] we removed these layers from both networks.
>
> We observed a similar behavior with these networks to the one seen with the LeNet backbone. Namely, the bound is data-dependent and gives non-vacuous guarantees. Since NeurIPS guidelines forbid adding files or external links to the review, we report here the mean difference between the empirical bound and the test error in the table below. The full results, including figures similar to Figure 2 in the paper, will be added to the revised version of the paper.
>
> Network | 0.25 | 0.5 | 0.75 | 1.0
> ----------|--------|--------|--------|--------
> ResNet  | 0.24 | 0.17 | 0.14  | 0.12
> MobileNetV2  |  0.21  |  0.17  |  0.15  |  0.13
>
> **5**
> > “Furthermore, there are some approximations involved in computing the bound [...]. It is a bit unclear how finite-sample estimates of these quantities affect the actual bound and, ideally, they should be taken into account when providing the actual guarantee.“
>
> This is true. At the moment we are unable to bound the MC error, but can only estimate it using methods such as jackknife that are also MC estimates. We will revisit the text to make sure it is clear that we only estimate the bound.
>
> **6**
> > “The experimental section and setup is a bit unclear with respect to a couple of important FL training hyperparameters. In prior work [2], it has been shown that in such scenarios, doing less epochs of local optimization (denoted as E) and using more clients per round can be beneficial for the final performance of the model when using FedAvg [...]. Going over the authors’ appendix, it seems that for the baselines they use E≈20 but E∈set(1,3) for pFedGP.“
>
> We thank the reviewer for referring us to Hsu et al. (2019). We agree with the reviewer that the number of local optimization steps and the number of clients to use in order to perform updates is an important topic. For a fair comparison we fixed the number of clients to 5, which is common in the FL literature (e.g., [6]) for all baselines besides pFedHN which applies updates to one client at each time by design. So we could expect all methods to gain benefit as a result of optimizing this hyper-parameter. Regarding the number of local optimization steps, we would like to note that there was a mistake in the appendix stating that we ran the baseline methods for 20 epochs. Since the training procedure of baseline methods varies significantly, we ran the baselines FOLA, LG-FedAvg, pFedMe, FedU and pFedHN according to the recommended configuration in their papers or code (which is usually a few epochs). Regarding FedAvg, FedPer and pFedMe we used the setting proposed by pFedME [6] with 20 *iterations* (not *epochs*) per client. An iteration corresponds to a batch size of 64 examples. Therefore, for example, in CIFAR-10 with 50/100/500 clients, 20 iterations actually corresponds to ~2/6/16 epochs. When considering the number of communication rounds (1000), on expectation, each client performed a total of 40/60/32 epochs correspondingly.
>
> Following the reviewer comment we applied a grid search over the number of iterations in {10, 20, 40, 80} for the baselines FedAvg, pFedMe, pFedHN on CIFAR-10 and CIFAR-100. We also searched for personal learning rates in {5e-2, 1e-2, 5e-3, 1e-3}. We report here the test accuracy for 50 and 100 clients in the table below (selection was done based on the validation set). Comparing this table to Table 1 in the paper shows similar accuracies regardless of the number of iterations being used. We hope that these results and our explanation eliminate any concern regarding the legitimacy of our experiments.
>
> . | CIFAR-10 | CIFAR-10 | CIFAR-100 | CIFAR-100
> ----------|--------|--------|--------|--------
> Num. clients |	50 |    100 |  50  |    100
> FedAvg | 58.9±1.3 | 59.0±0.6 | 23.7±0.9  | 24.2±0.7
> pFedMe |  85.3±0.2   |  84.1±0.4  |  52.5±1.4  |  51.2±0.6
> pFedHn | **89.6±0.3** | 87.0±0.3 | 60.05±0.7 | 50.6±0.3
> pFedGP | 89.2±0.3 | **88.8±0.2** | **63.3±0.1** | **61.3±0.2**

---

> > ### Author Response · Authors · 2021-08-10
> > **Response to Reviewer NyvV - Part 2**
> >
> > **7**
> > > “In fact, the reference [2] has managed to train a Lenet type of network with FedAvg on a similar non-iid version of CIFAR 10 and got up to ~73% performance, which is much better than the ~60% this work reports. It would be good if the authors adapt the hyperparameters appropriately so that the comparison is more meaningful.“
> >
> > Indeed in (Hsu et al., 2019) FedAvg achieves an accuracy of ~73% while in our work it obtains ~60% on CIFAR-10 with 100 clients. The reason is that the setting is quite different. In (Hsu et al., 2019) the ~73% accuracy was obtained using a data partition according to a dirichlet distribution, and was tested on a uniform distribution. Our data partition scheme is more like the sort-and-parition method that was mentioned in this study, and each client's test distribution matches its train distribution. Hsu et al., (2019) did not present results for this setting. In fact, in [7, 8], which seemed to use a more similar training procedure to ours, FedAvg obtains a lower accuracy than the reported accuracy in our paper.
> >
> > **8**
> > > “Finally, for a better personalised model performance baseline that is quite simple you should consider the performance of the locally fine-tuned model when training with FedAvg, see [3].“
> >
> > Thank you for referring us to (Jiang et al., 2019). We will add a discussion about it and similar studies in the revised version of the paper. Our impression is that this federated learning approach is based on the MAML algorithm. Following Reviewer *anUe* suggestion, which also referred us to a MAML-based approach [9], and due to time constraints, we compared our method to [9] only since it is more recent and similar to our learning procedure. We present the test accuracy of this method on CIFAR-10 and CIFAR-100 with 50 and 100 clients in the table below. Comparing this method to pFedGP reveals that our approach outperforms this baseline as well.
> >
> > . | CIFAR-10 | CIFAR-10 | CIFAR-100 | CIFAR-100
> > ----------|--------|--------|--------|--------
> > Num. clients |	50 |    100 |  50  |    100
> > Per-FedAvg [9] | 83.6±0.5 | 82.3±0.2 | 45.4±0.8  | 47.9±0.7
> > pFedGP | **89.2±0.3** | **88.8±0.2** | **63.3±0.1** | **61.3±0.2**
> >
> > **9**
> > > “One result that was relatively surprising was that FedPer had much worse performance than pFedGP, as they are quite similar in a sense; they both use the same backbone, except that one uses a linear classifier whereas the other uses a GP. It would be great if the authors can explain why this is the case.“
> >
> > We do not view GPs as equivalent to a linear layer. GPs with RBF kernels (as we used in our work) correspond to an infinite basis function representation which is effectively a hidden layer with an infinite number of hidden units [10]. Therefore, discriminative-wise it is much more powerful than a linear classifier. Also, when the dataset is limited in size, even a linear layer can overfit while GPs, which are Bayesian learners, don’t overfit as easily. Hence, the performance gap between the models.
> >
> > **10**
> > > “Furthermore, upon inspecting the author’s code it seems that they use two convolutional followed by two fully connected layers + one extra embedding layer, before feeding the embeddings to the GP. This is a larger model than the traditional lenet [...]. What kind of backbone did the authors use for their baselines?”
> >
> > We used *exactly* the same backbone for all methods. Indeed we used a LeNet based architecture which has two convolution layers, two non-linear fully connected layers and another linear layer. This is inline with studies in this field that use a network with two convolution layers, and two non-linear fully connected layers [7, 8, 11]. We added an affine layer to avoid zero entries in the last layer (as a result from using ReLU activations) for models that use this layer in their personal model, such as pFedGP and FedPer.
> >
> > **11**
> > > “At line 192 in the main text the authors mention that inducing points can distort the true class distribution for pFedGP and that they use reweighting to combat this effect. [...] I believe that the authors should provide an example of this event and why it happens. [...] what is q(y,x), p(y,x)?”
> >
> > Thank you for this point. We will further elaborate on it in the revised version of the paper.
> > We defined the inducing inputs globally (at the server) and evenly among the classes (100 per class). Therefore, when constructing the GP per node we changed the original (true) data class distribution (it was made uniform). As a result, for a new example the classifier is now more likely to predict classes that had less representation in the original dataset than it should have been if the GP was constructed according to the true class distribution. As an example, consider a binary classification problem having 90 examples from the first class and 10 examples from the second class (therefore, $q(y=1) = 0.9$, and $q(y=2) = 0.1$). Assume we defined 50 inducing inputs per class, so now during test time the model sees (90 + 50 = ) 140 samples from the first class and (10 + 50 = ) 60 from the second which corresponds to probabilities $p(y=1)=0.7$ and $p(y=2)=0.3$.
> >
> > Applying the correction effectively reduces the likelihood of predicting from the second class. This correction does not impact when the class distribution per client is even. However, when the class distribution is skewed (such as in the experiments in Section 6.3) it improves the model accuracy. We will show an ablation study for this correction in the final version.
> >
> > **12**
> > > “On lines 301-302 the authors mention that pFedGP-data is especially helpful when few datapoints per class are available, nevertheless, this seems to be the case only for CIFAR100 but not for CIFAR10 / CINIC-10. It would be helpful in the authors explain why.”
> >
> > We agree with the reviewer that pFedGP-IP-data was more beneficial on CIFAR-100 with 500 clients. The main reason is that the number of classes in the CIFAR-100 experiment is larger, so the number of data points per client per class is smaller. This point is further validated by an experiment shown in Appendix E.2. We tested our model performance on CIFAR-10 by varying the training dataset size and keeping the number of clients fixed. We show that pFedGP-IP-data outperforms pFedGP when there are 40 examples per client or less, which corresponds to 20 examples per class.
> >
> > **13**
> > > “Finally, it seems that the inducing points were fixed to 100 for all the experiments? How critical of a hyper parameter is the number of inducing points for the performance of pFedGP?”
> >
> > The reviewer is indeed correct. The number of inducing points was fixed to 100 because we believed it balances well between the computational requirements and the ability to capture the input space of the GP. In Appendix E.5 we present an ablation that tested the effect of the number of inducing points. We found that the number of inducing points can be reduced by almost half without any degradation in the performance. When taking only 20% of the inducing points the accuracy dropped by ~1.3% which is relatively small compared to the gain in computational complexity. Following this question we also tested the impact of the number of inducing points in CIFAR-10 and CIFAR-100 with 50 and 100 clients as well. We saw a similar behavior on these datasets. In fact, when we took only 20% of the inducing points the accuracy drop was ~0.7% which is even smaller than the one seen in CINIC-10.
> >
> > [1] Cai, D., Wang, Q., Liu, Y., Liu, Y., Wang, S., & Xu, M. (2021, June). Towards Ubiquitous Learning: A First Measurement of On-Device Training Performance. In Proceedings of the 5th International Workshop on Embedded and Mobile Deep Learning (pp. 31-36).
> >
> > [2] Dong, K., Eriksson, D., Nickisch, H., Bindel, D., & Wilson, A. G. (2017, December). Scalable log determinants for Gaussian process kernel learning. In Proceedings of the 31st International Conference on Neural Information Processing Systems (pp. 6330-6340).
> >
> > [3] Wang, K., Pleiss, G., Gardner, J., Tyree, S., Weinberger, K. Q., & Wilson, A. G. (2019). Exact Gaussian processes on a million data points. Advances in Neural Information Processing Systems, 32, 14648-14659.
> >
> > [4] Yu, F., Zhang, W., Qin, Z., Xu, Z., Wang, D., Liu, C., Tian. Z, & Chen, X. (2020). Heterogeneous federated learning. arXiv preprint arXiv:2008.06767.
> >
> > [5] Li, Q., Diao, Y., Chen, Q., & He, B. (2021). Federated learning on non-iid data silos: An experimental study. arXiv preprint arXiv:2102.02079.
> >
> > [6] T Dinh, C., Tran, N., & Nguyen, T. D. (2020). Personalized Federated Learning with Moreau Envelopes. Advances in Neural Information Processing Systems, 33.
> >
> > [7] Hao, W., Mehta, N., Liang, K. J., Cheng, P., El-Khamy, M., & Carin, L. (2020). WAFFLe: Weight Anonymized Factorization for Federated Learning. arXiv preprint arXiv:2008.05687.
> >
> > [8] Shamsian, A., Navon, A., Fetaya, E., & Chechik, G. (2021). Personalized Federated Learning using Hypernetworks. ICML 2021.
> >
> > [9] Fallah, A., Mokhtari, A., & Ozdaglar, A. (2020). Personalized federated learning with theoretical guarantees: A model-agnostic meta-learning approach. Advances in Neural Information Processing Systems, 33, 3557-3568.
> >
> > [10] Wilson, A. G., Hu, Z., Salakhutdinov, R., & Xing, E. P. (2016, May). Deep kernel learning. In Artificial intelligence and statistics (pp. 370-378). PMLR.
> >
> > [11] McMahan, B., Moore, E., Ramage, D., Hampson, S., & y Arcas, B. A. (2017, April). Communication-efficient learning of deep networks from decentralized data. In Artificial intelligence and statistics (pp. 1273-1282). PMLR.

---

> > > ### Comment · Reviewer_NyvV · 2021-08-17
> > > **Response to author rebuttal**
> > >
> > > I would like to thank the authors for the extensive rebuttal that clears most of my concerns. I will thus  increase my score to a 6, as the results are now stronger and more convincing. I highly encourage the authors to revise the submission with the additional results and also take into account the comments from all of the reviewers. One further comment from my side is that [11] uses a CNN with two convolutional layers, one fully connected layer mapping to 512 dimensions and a classification layer (thus it is different than the one the authors used) (see https://arxiv.org/pdf/1602.05629.pdf, section 3, second paragraph). [7] seems to also follow [11] in terms of the architecture used. Nevertheless, this is fine and does not affect the results, as the authors did all of the baseline experiments with the same CNN that they employed for pFedGP.

---

> > > > ### Author Response · Authors · 2021-08-17
> > > > **Response to Reviewer NyvV Comment**
> > > >
> > > > We thank the reviewer for revaluating the paper and for increasing the score based on our response. We highly appreciate that. We will take into account all of the comments and suggestions that were raised by the reviewers and revise the paper accordingly.

---

### Official Review · Reviewer_ys9C · 2021-07-16

**Rating:** 6
**Confidence:** 4

**Summary:**

This paper proposes to use Gaussian processes for personalized federated learning (PFL), and use deep kernel learning to allow all clients to jointly learn a shared kernel. Two inducing points-based extensions are also introduced to facilitate more information sharing and reduce the computational cost respectively.

**Limitations And Societal Impact:**

Yes.

**Main Review:**

The paper proposes an interesting application of GP with deep kernel learning to federated learning, and has shown competitive empirical performances for the proposed algorithm. My main concern about the current paper is regarding the technical novelty, in particular, most of the technical contributions seem to adapted results from existing GP methods, and the proposed methods seem like pure applications of GP to FL.

Strengths:
- In the experiments, a comprehensive set of existing methods have been compared with. The experimental results in Table 1 are very compelling, and the results in Figure 3 also look very nice, clearly demonstrating the advantage of GP in terms of uncertainty calibration.

Weaknesses/Questions:
- I think the use of GP for PFL is not well motivated. I think the essence of Personalized FL is to incorporate some amount of personalized information (learned from a client's local data) into a jointly learned ML model. In this sense, GP is only one of many possible choices. In fact, the most natural choice in my opinion is to use a mixture of global and local models, which has also been mentioned in related works. So I think a better and clearer motivation is needed.
- My overall impression is that the technical contents of this paper is almost exclusively about GP, rather than FL. For example, in Section 4.1, the only place about FL is the paragraph starting from line 149 where we can use standard FL to learn a neural network which is used to extract representation for kernel learning; in fact, all equations in this section are from existing GP techniques if I understood correctly. Sections 4.2 and 4.3 simply treat the inducing inputs and outputs as additional parameters to be shared among clients (again the equations such as equation 6 and 7 in these two sections are from existing results on GP), and the generalization bound in Section 5 is a pure GP result. Therefore, in my opinion, this paper is an application paper which applies existing results on GP (with some modifications) to FL, which hence may limit the technical contributions.
- Section 4.2: how do you learn the inducing points? The method in this section uses inducing points as the parameters for information sharing among different clients to overcome the nonparametric nature of GPs, so the paper [1] listed below should be cited here which uses another common technique (random Fourier features) for the same purpose.
- Section 4.3: lines 208-209, the inducing points are also defined globally by the server, so it's unclear to me what's the difference between the methods in Sections 4.2 and 4.3?
- Section 5: As mentioned above, the results in this section seem to have nothing to do with FL. Therefore, I doubt whether it will add much value to the current work as a paper for PFL.
- Section 5: just to clarify, is it correct that Theorem 1 is an existing result from [56], and your contribution here is to calculate the upper bound for your specific choice of GP classifier with Polya-Gamma augmentation? Lines 248-249: why have you chosen this approximation for $Q(\boldsymbol{\omega}_i)$? Is it like a marginalization over the $\boldsymbol{f}_i$'s?
- Experiments, is there a way to quantify and compare your computational cost with other methods? Because GPs are usually associated with large computational cost compared with parametric models like neural networks. One of the baselines is "Local" with uses a GP together with a locally learned kernel network, so have you tested the performance of standard GP (without kernel learning)? Not sure if I understood correctly, in line 292, it is mentioned that for CIFAR-10, every agent only has access to data belonging to two classes; however, in Table 1, how can the Local method still achieve an accuracy of above 70%? The same applies to the other two datasets.
- The paragraph starting from line 127: I feel that more technical background on GP-Tree is needed here since it's the backbone for all algorithms in this work.

[1] Federated Bayesian optimization via Thompson sampling. Dai, et. al. NeurIPS 2020.

Some minor comments:
- line 119: one of the $\mathbf{x}_i$'s inside exp should be $\mathbf{x}_j$
- line 269: "federate" -> "federated"

**Time Spent Reviewing:**

6 hours

---

> ### Author Response · Authors · 2021-08-10
> **Response to Reviewer ys9C - Part 1**
>
> We thank the reviewer for the thoughtful comments. Please see our responses below.
>
> **1**
> > “I think the use of GP for PFL is not well motivated. [...] In this sense, GP is only one of many possible choices. [...]. In fact, the most natural choice in my opinion is to use a mixture of global and local models [...].”
>
> Thank you for pointing that out. We will motivate better why GPs are a promising direction for PFL and not just one of many possibilities. PFL introduces some unique challenges: (i) the amount of data per client is limited [1], otherwise pure local training would have been sufficient; (ii) the input distribution can vary significantly between clients; and (iii) often real-world federated learning systems are deployed in safety critical systems which require calibrated predictions (e.g., [2, 3]). GP models are well suited for all of these challenges. They perform well on limited data due to their Bayesian nature and they provide well-calibrated predictions due to accurate uncertainty estimation. To use GPs on structured data such as images a common solution is to apply deep kernel learning which learns an expressive kernel function; however, when the amount of data is limited they can severely overfit, thus negating the motivation to use them in the first place. Therefore, we suggested overcoming this limitation by learning a shared kernel among all clients. With this model design we can gain from the GP benefits and avoid their limitations. As evidence, pFedGP shows superior performance to the baselines, even though some are using a mixture of global and local models.
>
> **2**
> > “My overall impression is that the technical contents of this paper is almost exclusively about GP, rather than FL. [...] in fact, all equations in this section are from existing GP techniques [...]. Therefore, in my opinion, this paper is an application paper which applies existing results on GP (with some modifications) to FL, which hence may limit the technical contributions.”
>
> Indeed in the paper we applied existing GP techniques, with modifications, to PFL. While the majority of the text is about GP, as it is the main algorithmic tool we used, we showed how to adjust existing methods so they will work well under the PFL setting. We contributed conceptual ideas, such as the shared learning of the kernel and the sharing of the inducing inputs that may be applied to any GP classifier in this setting. Therefore, we kindly disagree with the claim that the paper is exclusively about GPs, but rather we believe that it is on how to make GPs work in the PFL setting. Furthermore, we view the modifications made as non-trivial and believe that they include technical contributions. Therefore, we do not think our paper lacks technical significance.
>
> To obtain the results in the paper we introduced several novelties that, to the best of our knowledge, weren’t presented before:
> 1. We presented an alternative learning scheme of the model parameters with GP-Tree based on a Gibbs sampling approach that is more flexible and is able to quantify uncertainty more reliably.
> 2. To the best of our knowledge, we are the first to propose the inducing point method in section 4.2 and the class distribution correction for it.
> 3. We are the first to show how FITC can be combined with a Polya-Gamma based classifier to obtain a more scalable classifier and how to learn with it the model parameters using MCMC samples for classification tasks.
> 4. We were also the first to adapt the PAC Bayes bound to Polya-Gamma classifiers and provide an estimation of the bound. Although this result is general it is very relevant to the scenario of having a new client entering the system.
>
> We believe that these novelties accumulate to a novel, general, and complete solution for the FL setup. As an indication, our method outperforms by a large margin other baselines, some of which may have larger technical content, in a variety of settings and on several datasets.
>
> **3**
> > “how do you learn the inducing points? [...] so the paper [1] listed below should be cited here [...]. it's unclear to me what's the difference between the methods in Sections 4.2 and 4.3?”
>
> Thanks for referring us to (Dai et al., 2020), we will address this study in the revised version of the paper. We will also make Sections 4.2 and 4.3 more clear. We will now explain the differences better here.
>
> In both the inducing point methods (presented in Sections 4.2 and 4.3) the inducing inputs are defined in the last feature space of the neural network and they are shared among clients. Also, in both methods the inducing locations are learned via gradient descent similarly to the NN parameters.
>
> In the method presented in Section 4.2 we use the full GP model formulas (Section 4.1) on the pseudo-data. During training, we regard *only* the set of inducing inputs-labels as the available data and use them to predict the clients training data. To learn the model parameters, we maximize Eq. 3 and use the gradient estimator in Eq. 4. During test time, we use both the inducing inputs-labels and the training data for posterior inference and apply predictions with the predictive likelihood for new data points. This increases computation during inference and aims to help predictions in the extreme low-data regime.
>
> In the method presented in Section 4.3 we used a different technique. The inducing points are treated differently than the training data, in a way similar to the FITC, for the purpose of allowing for faster computations.
>
> **4**
> > “Section 5: [...]  the results in this section seem to have nothing to do with FL. [...]. Just to clarify, is it correct that Theorem 1 is an existing result from [56], and your contribution here is to calculate the upper bound for your specific choice of GP classifier with Polya-Gamma augmentation?”
>
> Indeed Theorem 1 is an existing result from [56] and we adjusted it to GP classifiers with the Polya-Gamma augmentation. The results in this section have much to do with FL and for general distributed ML. Often in FL systems clients join the system after it was learned, which in our case is after the kernel is learned. The result in the paper is applicable to this case exactly, and is not applicable to the standard deep kernel learning setting. There is much benefit in a useful bound on the error of the model for these novel clients. Having a reliable estimate of the error may prevent a prohibitive retraining of the entire system.
>
> **5**
> > “Lines 248-249: why have you chosen this approximation for $Q(\omega_i)$? Is it like a marginalization over the $f_i$'s?”
>
> Yes. This is a Monte-Carlo estimation of the marginal distribution.
>
> **6**
> > “Experiments, is there a way to quantify and compare your computational cost with other methods?”
>
> Reviewer *NyvV* had a similar concern. We will share the same answers for both. pFedGP is indeed more computational intensive compared to standard methods such as FedAvg. In return to that additional complexity we were able to obtain substantial performance gains compared to the baseline methods. When implementing pFedGP there are some trade-offs, for example, iterating over the tree can be either sequential to obtain lower memory cost or parallelized to obtain shorter running times. When applying the Gibbs sampling, the complexity can be affected by the number of parallel Gibbs chains and the number of MCMC steps which constitute a tradeoff between accuracy and runtime. In fact, in our experiments we noticed a reasonable drop in accuracy (~2% at most) when using only one chain and taking a few MCMC steps.
>
> Regarding the clustering algorithm, since the k-means algorithm is applied on the class prototypes only (one representative example for each class), the number of data points is small and in the worst case is at the size of the number of classes.
>
> Following this comment, we tracked pFedGP and baseline methods memory usage and runtime on CIFAR-10 and CIFAR-100 with 50 clients. For comparability, we fixed the number of epochs that each sampled client makes to one. pFedGP needed ~1.4/1.6 GB memory, and a run-time of ~1/2 hours for CIFAR-10/100 correspondly. Baseline methods needed 1.1-1.3 GB memory and took ~25 minutes to run on both datasets. We believe pFedGP computational requirements are reasonable for running it in current FL systems [4] and considering that the code is highly not optimized.
>
> Also, when considering only the inference time we can obtain additional speed up by precomputing and caching many of the computations (e.g. Cholesky decomposition). Baseline method needed ~0.6ms to run a single example while pFedGP needed ~6ms for CIFAR-10 and ~30ms for CIFAR-100. While relatively much slower, it is still quite fast and can run in real-time.
>
> Finally, we note that while scaling GPs is an open research problem [5, 6] and many advances can be used to scale our method, our main focus is the low data regime where these issues are not as restrictive.

---

> > ### Author Response · Authors · 2021-08-10
> > **Response to Reviewer ys9C - Part 2**
> >
> > **7**
> > > “One of the baselines is "Local" [...] have you tested the performance of standard GP (without kernel learning)? [...] for CIFAR-10, every agent only has access to data belonging to two classes; [...] how can the Local method still achieve an accuracy of above 70%? The same applies to the other two datasets.”
> >
> > The local model uses a personal NN for each client and applies kernel learning with it. We did not test standard GP without kernel learning since common kernels do not provide a good measure of similarity for structured data such as images. Kernel learning helps to alleviate this issue.
> >
> > Regarding the accuracy of the local model, please note that the test distribution at each client is the same as its training distribution. Therefore, in CIFAR-10 experiments the test set for each client is made of the same two classes that appeared in the training set. Similarly, in CIFAR-100/CINIC-10 the same 10/4 classes that appeared in the training set appear in the test set of each client. Therefore, the obtained accuracies for CIFAR-10 are sensible as 50% is the trivial baseline accuracy.
> >
> > **8**
> > > “The paragraph starting from line 127: I feel that more technical background on GP-Tree is needed here since it's the backbone for all algorithms in this work.”
> >
> > We agree. Due to space limitation we could not elaborate on GP-Tree in the main text. We provided an extensive background for this method in Appendix A.2.
> >
> > **9**
> > > “Some minor comments: [...]”
> >
> > Thanks a lot!
> >
> >
> > [1] Zhang, C., Xie, Y., Bai, H., Yu, B., Li, W., & Gao, Y. (2021). A survey on federated learning. Knowledge-Based Systems, 216, 106775.
> >
> > [2] Lu, S., Yao, Y., & Shi, W. (2019). Collaborative learning on the edges: A case study on connected vehicles. In 2nd {USENIX} Workshop on Hot Topics in Edge Computing (HotEdge 19).
> >
> > [3] Huang, L., Shea, A. L., Qian, H., Masurkar, A., Deng, H., & Liu, D. (2019). Patient clustering improves efficiency of federated machine learning to predict mortality and hospital stay time using distributed electronic medical records. Journal of biomedical informatics, 99, 103291.
> >
> > [4] Cai, D., Wang, Q., Liu, Y., Liu, Y., Wang, S., & Xu, M. (2021, June). Towards Ubiquitous Learning: A First Measurement of On-Device Training Performance. In Proceedings of the 5th International Workshop on Embedded and Mobile Deep Learning (pp. 31-36).
> >
> > [5] Dong, K., Eriksson, D., Nickisch, H., Bindel, D., & Wilson, A. G. (2017, December). Scalable log determinants for Gaussian process kernel learning. In Proceedings of the 31st International Conference on Neural Information Processing Systems (pp. 6330-6340).
> >
> > [6] Wang, K., Pleiss, G., Gardner, J., Tyree, S., Weinberger, K. Q., & Wilson, A. G. (2019). Exact Gaussian processes on a million data points. Advances in Neural Information Processing Systems, 32, 14648-14659.

---

> > > ### Comment · Reviewer_ys9C · 2021-08-23
> > > **Response to Rebuttal**
> > >
> > > I appreciate the authors' rebuttal. The responses cleared my previous major concerns. In particular, points 2 and 6 in the responses are particularly useful and I suggest the authors include these two points in the revised paper. I'm still not fully convinced by point 4 in the response (about the theoretical results in Section 5), but this doesn't take away the empirical contribution of this paper. Hence, I've increased my score to 6.

---

> > > > ### Author Response · Authors · 2021-08-23
> > > > **Response to Reviewer ys9C Comment**
> > > >
> > > > We thank the reviewer for revaluating the paper and for increasing the score. We greatly appreciate that. We will include points 2 & 6 as well as other comments that were raised in the revised version of the paper.

---

### Author Response · Authors · 2021-08-10
**General Comment**

We thank the reviewers for their time and effort and their constructive and insightful feedback. We are encouraged that the reviewers found our approach *interesting* (R-ys9C), *novel* (R-anUe), and *clear* (R-NyvV). Reviewers were *impressed* by our method results (R-ys9C, R-anUe) and *appreciated* the ability of it to quantify uncertainty (R-NyvV). The main concerns were with the technical novelty of the paper (R-ys9C, R-anUe, R-NyvV), the computational cost (R-ys9C, R-NyvV), and the experimental setting (R-NyvV). We address all issues and concerns in our responses below.

---

### Decision · Program_Chairs · 2021-09-27

**Decision:**

Accept (Poster)

**Comment:**

The paper adopts and adapts a number of known GP learning techniques to develop a method for personalised federated learning. The proposed method is computationally intensive, but performs well in practice and even comes with theoretical utility analysis. After author feedback, all reviewers appear happy with the paper and recommend acceptance, assuming the authors revise the paper to incorporate the material from the author feedback.